# Judgment Operators: A Composition-Invariant Substrate for Multi-Agent Action Spaces

Jun Li [1]

## Abstract

As large language models (LLMs) are increasingly composed into heterogeneous multi-agent systems, a fundamental reliability challenge emerges: knowledge and governance **fragment** across agents, leading to composition-dependent behaviors and linear scaling of violations. Two empirical observations motivate our approach: internal correction methods (Reflexion, CRITIC, Self-Refine) fail to enforce execution-time constraints deterministically, and filter-based methods (LlamaGuard) sacrifice task utility without repair. We introduce **Judgment Operators (JO)**, a decision-time framework implementing four-way intervention semantics (ALLOW, EDIT, ESCALATE, DENY) via a portable artifact $J = (\mathcal{C}, \mathcal{P})$, enabling minimal repair without modifying agent internals. We establish: (1) **composition-invariant enforcement** with constant violation probability, motivated by a necessity lower bound showing no decentralized scheme achieves $O(1)$ violation probability; (2) **sublinear** mistake accumulation via JO-A; and (3) **semantic preservation** for code transformation. Programmatically verifiable constraints provide the formal foundation; empirically, robustness extends beyond this scope (partial mediation 3–5× better than theoretical worst-case). JO achieves 0% observed violation rate (vs. 48–68% baselines), 13.5–20.5% zero-shot cross-model transfer where few-shot prompting fails, and provides a **portable, auditable, and composable** interface for governance and capability injection in multi-agent LLM systems.

[1]Veyra, veyraintelligence.com, San Francisco, CA. Correspondence to: Jun Li <jun@veyraintelligence.com>.

*Proceedings of the 43$^{rd}$ International Conference on Machine Learning*, Seoul, South Korea. PMLR 306, 2026. Copyright 2026 by the author(s).

## 1. Introduction

### 1.1. Problem: Fragmentation with Linear Scaling

As large language models (LLMs) are composed into multi-agent systems, both capability and governance fragment across agents. Expertise (e.g., API usage) and norms (e.g., admissibility) become embedded in individual parameters, prompts, or non-transferable training data, leading to composition-dependent behaviors. Critically, violations scale *linearly* with agent count: for $n$ agents each violating with probability $\epsilon$, the system-wide violation probability is $P_{\text{viol}} = O(n\epsilon)$, undermining reliability. We focus on programmatically verifiable constraints (e.g., formats, schemas, state logic), which admit rigorous analysis and provide a concrete foundation for governance.

### 1.2. Key Insight: Decision-Time Projection Interface

Existing approaches internalize governance and capability within agents via fine-tuning, prompting, or learned critics, causing fragmentation and requiring retraining. Even centralized prompting reduces violations only partially (only 2% task success under our step budget) and fails to guarantee execution.

We externalize governance and capability into a decision-time projection. A central **Judgment Operator** $\Pi_J$ separates action *proposal* from *judgment*, projecting actions onto an admissible set $\mathcal{X}_J$ using an artifact $J = (\mathcal{C}, \mathcal{P})$, where $\mathcal{C}$ encodes constraints and $\mathcal{P}$ encodes corrective precedents $(x, x')$ with $x' \in \mathcal{X}_J$. The operator implements **four-way intervention semantics**: $\Pi_J(x) \in \{\text{ALLOW}(x'), \text{EDIT}(x'), \text{ESCALATE}, \text{DENY}\}$, where EDIT corrects using precedents (capability injection) and ESCALATE handles ambiguity (governance oversight).

**JO-A: Online Nonparametric Learning.** We introduce JO-A, which learns these projections online: precedents $\mathcal{P}$ serve as training data for the projection map $x \to x'$, with retrieval implementing nearest-neighbor regression. This yields a nonparametric, decision-time learning mechanism without retraining agents.

## 1.3. Guarantees and Empirical Validation

We establish three guarantees: (1) Composition-invariant enforcement, where violation probability is constant in agent count (bounded by checker error $\epsilon_J$), versus $O(n\epsilon)$ linear scaling without JO; (2) learnable correction under recurrence, yielding sublinear violation growth over time; and (3) structure-preserving projection, enabling safe transfer of corrective knowledge across tasks and models.

Empirically, JO functions as a unified decision-time knowledge interface: it achieves zero observed violations in fully verifiable settings (versus 48–68% for baseline methods, and only 2% success for centralized prompting), improves task success by 9–11% under recurring failures, and enables 13.5–20.5% absolute zero-shot cross-model transfer where few-shot prompting yields no improvement.

## 1.4. Contributions and Impact

Our contributions are: **(1) Judgment Operators.** A decision-time projection interface that externalizes governance and corrective capability into portable artifacts. **(2) Composition-Invariance.** A formal guarantee converting linear violation scaling to a constant bound under complete mediation. **(3) Learnable Repair.** A precedent-based mechanism enabling reusable correction under recurring violations with sublinear mistake growth. **(4) Empirical Evidence.** Zero observed violations in fully verifiable settings, learning-based success gains, and zero-shot cross-model transfer across models.

**Toward a New Paradigm.** Judgment Operators offer a new paradigm for multi-agent systems: policies, norms, and knowledge become explicit artifacts that can be composed, audited, and updated globally, addressing fragmentation through **learnable composition-invariant projection**.

## 2. Related Work

### 2.1. Multi-Agent LLM Systems

LLMs enable autonomous agents via ReAct (Yao et al., 2023), planner-executor-critic architectures, and tool augmentation. Benchmarks (WebArena (Zhou et al., 2023), BrowserGym (de Chezelles et al., 2025), VIMA (Jiang et al., 2022), ScreenAgent (Niu et al., 2024)) show agent count amplifies violations—the linear scaling JO addresses. Multi-agent frameworks improve coordination (Sun et al., 2024; Zhang et al., 2025); debate mechanisms (Michael et al., 2023) increase deliberation but lack uniform enforcement. In MARL, QMIX (Rashid et al., 2018) and COMA (Foerster et al., 2018) achieve composition invariance via learned value functions; JO targets language agents at decision time without value learning.

### 2.2. Training-Time Alignment

RLHF, DPO, and RLVR (Wen et al., 2025) encode preferences at training time, entangling constraints with parameters. Related approaches include GRPO (Shao et al., 2024) and self-play (Chen et al., 2025). Prompt-based specification requires duplication and suffers inconsistent interpretation. JO enforces constraints *externally at decision time*, decoupled from training.

### 2.3. Runtime Governance and Guardrails

Governance frameworks (GaaS) decouple enforcement via declarative policies and runtime mediation without modifying agent internals (Gaurav et al., 2025), but typically act as evaluators without executable repair. Production guardrails reflect this gap: NeMo Guardrails (Rebedea et al., 2023) and Llama Guard (Inan et al., 2023) classify or reject violations, while Guardrails.ai re-prompts on validation failure without convergence guarantees. JO instead treats governance as a *decision-time projection problem*, minimally deforming inadmissible actions via precedents to ensure admissibility; unlike rejection-based systems, JO supports escalation as a first-class outcome while learning executable repairs (empirical comparison in Table 8).

### 2.4. Safety Shields and Constrained Decoding

Safe RL shields motivate action correction (Alshiekh et al., 2018; Fisac et al., 2019): shields synthesize LTL constraints that intervene on violations and project actions to nearest safe alternatives via reachability. LLM extensions such as AgentSpec (Wang et al., 2025) and ProbGuard (Wang et al., 2026) enforce runtime constraints but do not learn reusable repairs. Constrained decoding (Outlines (Willard & Louf, 2023), Guidance (Lundberg et al., 2023)) enforces syntax during generation but cannot handle semantic or stateful constraints. JO complements both by enforcing semantic constraints at execution time. We use *decision-time* to distinguish this from token-level inference-time decoding (empirical comparison in Table 6).

### 2.5. Decision-Time Correction

Traditional actor–critic methods learn critics during training. Internal critics (Reflexion (Shinn et al., 2023), CRITIC (Gou et al., 2023), Self-Refine (Madaan et al., 2023)) use internal reasoning loops, while structured planning embeds constraints in planners. JO's precedent retrieval relates to case-based reasoning and RAG, but retrieves *admissible repairs* rather than knowledge. The EDIT operation connects to program repair (SemFix (Nguyen et al., 2013)) and edit automata (Ligatti et al., 2005), which enforce via suppression or insertion. Unlike these, JO learns reusable corrections and constrains *executed actions* via external projection

*Table 1.* Contrasting Judgment Operators with Related Approaches

| | Post-hoc Moderator | Safety Shields | Internal Critics | Prompt-Embedde | JO |
|---|---|---|---|---|---|
| **Timing** | Post-gen. | Pre-exec. | Post-fail | Gen.-time | **Pre-exec.** |
| **Repair** | Reject | Safe action | Re-gen | Implicit | **Precedent** |
| **Portable** | Model-dep. | Policy-dep. | Prompt-dep. | Non-port. | **Artifact** |
| **Memory** | None | None | Trajectory | None | **Precedent** |

(empirical comparison in Table 8).

**Summary.** JO uniquely combines decision-time projection, portable artifacts, corrective editing, and composition invariance without training (Table 1).

## 3. Method

### 3.1. Problem Formulation: Multi-Agent Judgment Under Fragmentation

We consider a multi-agent system $\mathcal{A} = \{a_1, \ldots, a_n\}$ where each agent $a_i$ proposes actions $x_i \in \mathcal{X}$ (natural language plans, structured commands, or tool invocations), subject to programmatically verifiable constraints defining an **admissible set** $\mathcal{X}_J \subseteq \mathcal{X}$.

**Design principle.** We adopt *agents propose, the operator disposes*: all proposals are mediated by a centralized judgment operator $\Pi_J$ prior to execution, separating agent capabilities (proposal) from shared knowledge (admissibility) and preventing fragmentation at the architectural level.

### 3.2. Judgment Operators as Learnable Projections

**Judgment artifact.** The core of JO is a portable **judgment artifact** $J = (\mathcal{C}, \mathcal{P})$, where $\mathcal{C}$ specifies constraints via $\mathcal{X}_J = \{x \mid \forall c \in \mathcal{C}, c(x) = \text{true}\}$, and $\mathcal{P}$ stores precedents $(x, x')$ with $x' \in \mathcal{X}_J$ denoting admissible repairs.

**Geometric Optimization Perspective.** JO selects an admissible action by solving a decision-time optimization problem:

$$\Pi_J(x) = \arg\min_{x' \in \mathcal{X}_J} \underbrace{\ell(x', x)}_{\text{intervention cost}} + \lambda \cdot \underbrace{\Omega_J(x'; \Theta)}_{\text{artifact penalty}} \quad (1)$$

where $\ell$ measures deviation from the original proposal (e.g., token edit distance), $\Omega_J(\cdot; \Theta)$ encodes preferences from precedents $\mathcal{P}$, and $\lambda$ balances minimal intervention against learned corrections. This frames JO as a **minimal-intervention selector** over admissible candidates, implemented at decision time without retraining agents.

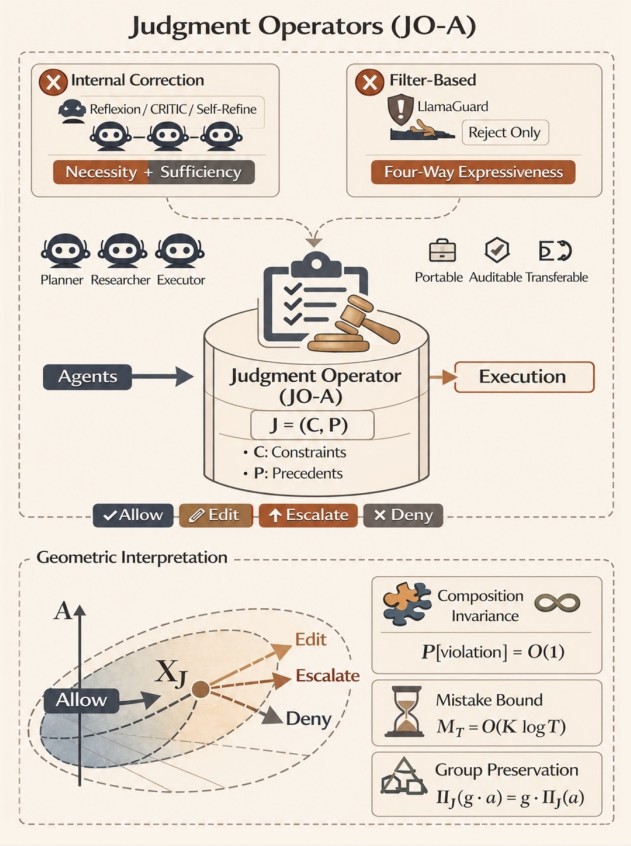

*Figure 1.* **JO as Multi-Agent Composition-Invariant Retraction.** JO defines a decision-time projection $\Pi_J : \mathcal{X} \to \mathcal{X}_J$ that retracts multi-agent proposals onto an admissible subspace via minimal surface edits, independent of agent composition.

**Artifact penalty.** The penalty $\Omega_J(x'; \Theta)$ encodes learned preferences over admissible candidates from artifact $J$, parameterized as a linear function of interpretable features:

$$\Omega_J(x'; \Theta) = \Theta^\top f(x'; J) \quad (2)$$

where $f(x'; J) \in \mathbb{R}^p$ extracts features from $x'$ relative to $J$:

- **Precedent support**: frequency of similar repairs in $\mathcal{P}$
- **Confidence**: similarity to nearest matching precedent
- **Edit severity**: magnitude of edits from the proposal
- **Constraint margin**: distance to boundaries in $\mathcal{C}$

The parameter vector $\Theta \in \mathbb{R}^p$ is updated online when violations occur, adjusting the weighting of these features based on observed failure patterns.

**Artifact-conditioned optimization and online adaptation.** Here "projection" denotes decision-time selection of a nearest admissible action via retrieval and minimal repair, not a convex operator. The artifact penalty $\Omega_J(\cdot; \Theta)$ is parameterized by a small set of artifact-level parameters $\Theta$ encoding preferences over admissible actions, such as precedent sup-

port and confidence. Action selection occurs at decision time via Eq. (1), while $\Theta$ is updated after execution when violations are observed. Upon a violation, JO-A applies a simple additive update

$$\Theta_{t+1} = \Theta_t + \alpha \cdot f(\tilde{x}_t; J), \qquad (3)$$

adapting admissibility preferences without modifying agent policies. Implementation details appear in Appendix B.3.

**Retract and Boundary View.** JO admits a natural interpretation as a *partial retract* onto the admissible set $\mathcal{X}_J$, i.e., a map $r : \mathcal{X} \to \mathcal{X}_J$ with $r(x) = x$ for all $x \in \mathcal{X}_J$. Operationally, ALLOW acts as the identity on $\mathcal{X}_J$, while EDIT applies a bounded deformation toward $\mathcal{X}_J$ when $x \notin \mathcal{X}_J$ but repairable; DENY and ESCALATE handle cases where a well-defined projection does not exist or is unstable. This characterizes JO as a *minimal-intervention* operator that preserves admissible structure while correcting violations (Appendix E.1).

**Boundary regimes and escalation.** ESCALATE captures boundary cases near $\partial \mathcal{X}_J$ where projection is ambiguous or unstable: multiple admissible repairs may exist, or small perturbations induce qualitatively different outcomes. In such regions, escalation prevents arbitrary projection, analogous to abstention in classification (Appendix E.2; empirical analysis in Appendix F.15).

**Practical Completeness of Quaternary Intervention.** The four actions {ALLOW, EDIT, ESCALATE, DENY} form a **practically complete** basis for decision-time intervention: they are *sufficient* to handle compliant actions, repairable violations, irreparable failures, and ambiguous cases requiring oversight, and *practically necessary* since removing any action compromises safety, utility, or auditability. A design-level justification is provided in Appendix E.3.

### 3.3. JO-A: Online Nonparametric Projection Learning

Algorithm 1 summarizes JO-A's decision-time adaptive projection. Unlike policy learning or retraining, JO-A adapts behavior at execution time by selecting among admissible candidates via Eq. (1). When violations recur, artifact-conditioned penalties are updated online (Eq. (3)), yielding sublinear violation growth under recurrence (Theorem 1; proof in Appendix D.1). A full pseudocode with audit logging, escalation handling, and artifact maintenance is provided in Appendix B.

**Implementation and complexity.** We implement Eq. (1) with token-level edit distance for $\ell$, an artifact penalty $\Omega_J(\cdot; \Theta)$ encoding precedent frequency and confidence, and $\lambda$ balancing minimal intervention against precedent guidance. The candidate set $\mathcal{S}(x_t)$ typically contains $< 10$ ele-

---

**Algorithm 1** Decision-Time Adaptive Projection (JO-A)

**Require:** Action proposal $x_t$, judgment artifact $J = (\mathcal{C}, \mathcal{P})$, penalty parameters $\Theta$
**Ensure:** Admissible action $x_t' \in \mathcal{S}_J \subseteq \mathcal{X}_J$ and intervention decision $d_t \in \{$ALLOW, EDIT, ESCALATE, DENY$\}$
1: **Step 1: Build candidate set** $\mathcal{S} \leftarrow \{x_t\} \cup$ RepairCandidates$(x_t, \mathcal{P})$
2: **Step 2: Hard constraints pruning** $\mathcal{S}_J \leftarrow \{x' \in \mathcal{S} \mid \forall c \in \mathcal{C}_{\text{hard}}, c(x') = \text{true}\}$
3: **if** ShouldEscalate$(x_t, \mathcal{C})$ **then**
4:     **return** (ESCALATE, NULL)
5: **end if**
6: **if** $\mathcal{S}_J = \emptyset$ **then**
7:     **return** (DENY, NULL)
8: **end if**
9: **Step 3: Select minimal-intervention action** $x_t' \leftarrow \arg\min_{x' \in \mathcal{S}_J} \ell(x', x_t) + \lambda \cdot \Omega_J(x'; \Theta)$
10: **Step 4: Determine decision type**
11: **if** $x_t' = x_t$ **then**
12:     $d_t \leftarrow$ ALLOW
13: **else**
14:     $d_t \leftarrow$ EDIT
15: **end if**
16: **Step 5: Online adaptation (post-execution)**
17: **if** violation observed with admissible repair $x_t^*$ **then**
18:     $\mathcal{P} \leftarrow \mathcal{P} \cup \{(x_t, x_t^*)\}$
19:     $\Theta \leftarrow \Theta + \alpha \cdot f(\tilde{x}_t; J)$ {Additive update penalizing violating feature profile}
20: **end if**
21: **return** $(d_t, x_t')$

---

ments, ensuring real-time feasibility even with $|\mathcal{P}| > 10^3$ precedents. Action embeddings $\phi$ use sentence embeddings (OpenAI text-embedding-3-small), achieving 87% top-1 retrieval accuracy on held-out violations. Per-step cost is $O(kL + md)$ for constraint checking and similarity search ($k$ constraints, $m$ precedents, $d$-dim embeddings), reducible to $O(\log m)$ with ANN (Faiss); empirical latency is reported in Appendix B.2. Full implementation details and hyperparameters appear in Appendix B.

**Online learning formulation.** JO-A learns the projection $P_J : \mathcal{X} \to \mathcal{X}_J$ online via nearest-neighbor regression. Given a proposal $x_t$, it retrieves the most similar precedent $(x_i, x_i') \in \mathcal{P}$ using $\text{sim}(\phi(x_t), \phi(x_i))$, where $\phi : \mathcal{X} \to \mathbb{R}^d$ is an embedding. The induced projection is

$$\hat{P}_J(x_t) = x_{i^*}', \qquad i^* = \arg\max_i \text{sim}(\phi(x_t), \phi(x_i)).$$

If no suitable precedent exists, JO-A falls back to rule-based repair or escalation. Upon a violation with admissible repair $x_t^*$, the artifact is updated $\mathcal{P} \leftarrow \mathcal{P} \cup \{(\phi(x_t), x_t^*)\}$, enabling adaptation under recurrence without retraining agents.

**Notation.** We use $\phi(x)$ to denote action embeddings and $\psi(s, x)$ for state–action features. In our implementation, $\psi(s_t, x_t) = [\phi(x_t); \phi(s_t)]$; retrieval uses

$\text{sim}(\phi(x_t), \phi(x_i))$, while state features are used only for constraint checking and repair selection.

**Learning the retract map.** JO-A learns a partial retract $r_J : \mathcal{X} \to \mathcal{X}_J$ online. Each precedent $(x_i, x_i') \in \mathcal{P}$ provides a sample with $x_i' = r_J(x_i)$. At decision time, the retract is approximated via nearest-neighbor regression, $\hat{r}_J(x) = x_{i*}'$ where $i^* = \arg\min_i d(x, x_i)$. This nonparametric formulation enables adaptation to recurring failures without retraining agents or inducing distribution shift.

**Approximation error.** The retract error decomposes as

$$d(\hat{r}_J(x), r_J(x)) \leq d(\hat{r}_J(x), r_J(x_{i*})) + d(r_J(x_{i*}), r_J(x)),$$

capturing retrieval and local approximation errors respectively; assumptions and bounds appear in Appendix E.1.

### 3.4. Theoretical Analysis

We establish three results: (1) centralized mediation is *necessary* for composition-invariant enforcement; (2) JO achieves a constant violation bound under explicit assumptions; and (3) JO-A exhibits sublinear violation growth under recurring failure modes. For code transformation, we provide an interpretive framework for semantic preservation.

**Theory-to-design narrative.** The theoretical contributions were motivated by two empirical observations, not derived abstractly. First, internal correction methods fail to enforce execution-time constraints deterministically (§4.4), motivating the necessity result and if-and-only-if characterization. Second, filter-based methods sacrifice task utility without repair (Table 8), motivating four-way expressive semantics. Empirical failures motivated the theoretical analysis, which in turn guided JO's design.

**Assumptions and scope.** Our results are **conditional guarantees** under the following assumptions: **(1) Complete mediation** (all actions pass through $\Pi_J$); **(2) Checker correctness** (false negative rate $\epsilon_J$ is small, empirically $\epsilon_J = 0.35\%$); **(3) Recurrence** (violations cluster into finitely many classes); and **(4) Repairability** (violations admit bounded edits to admissible actions).

We do not address adversarial sequences, partial mediation, or unverifiable constraints. The results characterize *architectural limits* of enforcement and identify where admissibility must be imposed to obtain composition invariance, rather than training-time learning guarantees.

### 3.4.1. NECESSITY OF CENTRALIZATION

**Linear scaling under decentralized enforcement.** For any decentralized scheme where agent $i$ independently enforces constraints with error probability $\epsilon_i$, the system-wide violation probability satisfies:

$$P_{\text{viol}}^{\text{decentralized}} \geq 1 - \prod_{i=1}^{n}(1 - \epsilon_i) = \Omega(n\epsilon) \qquad (4)$$

for homogeneous $\epsilon_i = \epsilon$ and small $\epsilon$.

*Proof.* The probability that no agent produces a violation is $\prod_{i=1}^{n}(1 - \epsilon_i)$, hence $P_{\text{viol}} = 1 - \prod_{i=1}^{n}(1 - \epsilon_i)$. For homogeneous $\epsilon_i = \epsilon$ and small $\epsilon$,

$$P_{\text{viol}} = 1 - (1 - \epsilon)^n \geq 1 - e^{-n\epsilon} \geq n\epsilon - (n\epsilon)^2/2 = \Omega(n\epsilon).$$

$\square$

*Implication:* Composition-invariant enforcement with $O(1)$ violation probability is **unattainable** by purely decentralized methods. This motivates JO's centralized architecture.

**Proposition 1** (JO Achieves Constant Bound under Complete Mediation)**.** *Under complete mediation and checker false negative rate $\epsilon_J$, JO ensures $P_{viol}^{JO} \leq \epsilon_J$, independent of agent count $n$.*

*Justification:* By construction, $\Pi_J$ only outputs actions in $\mathcal{X}_J$ (via ALLOW or EDIT) or blocks execution (via DENY or ESCALATE); violations arise only from checker false negatives, bounded by $\epsilon_J$. The guarantee is architectural and assumes complete mediation and checker coverage; blind spots or bypass attacks are not considered. Formal proofs are provided in Appendix D.2–D.3.

### 3.4.2. LEARNING FROM MISTAKES

We formalize when JO-A can *learn reusable repairs* from recurring violations.

**Recurrence and execution violations.** We assume violations cluster into a bounded number of recurring patterns: formally, a sequence has $(K, \rho)$-recurrence if violations fall into at most $K$ similarity classes under threshold $\rho$, capturing systematic failure modes rather than arbitrary noise (validated in Section 4.3). An execution violation occurs when the executed action $\tilde{x}_t \notin \mathcal{X}_J$, i.e., JO-A fails to project a violating proposal into the admissible set.

**Theorem 1** (Mistake Bound under Recurrence)**.** *Under $(K, \rho)$-recurrence with retrieval threshold $\tau \leq \rho$, JO-A incurs at most $M_T = O(K \log T)$ execution violations over $T$ steps, assuming monotone precedent coverage and bounded memory.*

*Proof sketch.* Each of the $K$ violation classes contributes at most one novel violation. Additional violations arise only when a covering precedent is evicted; standard cache analysis bounds this by $O(\log T)$ per class. The full proof appears in Appendix D.1.

*Comparison.* Static methods incur $M_T = O(T)$ violations; JO-A's sublinear bound formalizes learning from recurrence.

**Scope.** The bound applies to structured recurrence and is not adversarially robust: for non-recurring violations ($K = T$), JO-A reduces to static behavior.

**Proposition 2** (Active Precedent Selection). *Under $(K, \rho)$-recurrence, JO-A stores only violations whose similarity to existing precedents is below $\rho$, achieving the same $O(K \log T)$ mistake bound with $O(K)$ memory.*

*Interpretation.* Active selection prevents redundant storage of near-duplicate violations and aligns JO-A with sample-efficient online learning under recurrence.

### 3.4.3. SEMANTIC PRESERVATION

For domains with well-defined semantic equivalence, we provide an *interpretive* (non-universal) account of why JO-A preserves semantics (Appendix E.4).

**Definition 1** (Semantic Equivalence Group). *Let $G_{sem}$ denote semantics-preserving program transformations: $g \in G_{sem}$ iff $\text{exec}(g \cdot p, I) = \text{exec}(p, I)$ for all programs $p$ and inputs $I$. This includes $\alpha$-renaming, independent reordering, and formatting.*

**Proposition 3** (Semantic Preservation for Style Projection). *For code style tasks where $\mathcal{X}_J$ contains style-compliant programs, JO preserves semantics: for all $p$,*

$$\forall I : \text{exec}(\Pi_J^{style}(p), I) = \text{exec}(p, I), i.e., \Pi_J^{style}(p) \in [p]_{G_{sem}}.$$

*Validation.* All 47 programs pass their test suites before and after projection, with 0 AST tree-edit distance (Appendix F.10).

**Interpretive scope.** We do not claim a general equivariant projection theorem. The group-based view explains why JO-A preserves semantics when admissibility corresponds to $G_{sem}$-actions; it does not apply to purely syntactic or non-semantic constraints.

**Supporting results (sketch).** Under standard covering assumptions, nearest-neighbor projection yields

$$d(\hat{\Pi}_J(x), \Pi_J(x)) \leq \eta + L\epsilon.$$

For violation regions of doubling dimension $d$, $O(\epsilon^{-d})$ precedents suffice, implying error decay $O(|\mathcal{P}|^{-1/d})$ (Appendix D.4–D.5).

**Connections to ML frameworks.** JO connects to several ML paradigms while remaining distinct. It resembles safe RL and constrained optimization through decision-time projection (without training-time integration), imitation

learning via nonparametric reuse of admissible precedents, and nearest-neighbor regression with standard $O(1/\sqrt{n})$ approximation behavior. Unlike these approaches, JO provides explicit mistake bounds ($O(K \log T)$) and composition-invariant enforcement at execution time.

**Scope and practical robustness.** Programmatically verifiable constraints are required for formal guarantees—this is a mathematical necessity, not a practical limitation. Empirically, robustness extends beyond this scope: partial mediation performs 3–5$\times$ better than theoretical worst-case (Table 4), and Wikipedia/JSON results under partial checker coverage remain far below baseline (48–68%).

## 4. Experiments

### 4.1. Overview and Setup

We evaluate JO as a shared operator interface supporting (1) *governance enforcement* via execution-time verification and (2) *capability improvement* via reusable repair knowledge. We ask: (i) Can JO enforce governance across agents and models? (ii) Does JO acquire reusable corrective capability? (iii) Is external judgment fundamentally necessary?

We study three constraint regimes: **Format** (WebArena-Wikipedia), **Semantic** (Overcooked), and **Protocol** (JSON). For transfer and heterogeneous coordination studies, we use the full **WebArena-Wikipedia** setting; for recurrent-learning analyses, we use a **controlled Wikipedia**, a WebArena-style variant that fixes navigation to isolate recurring judgment and repair patterns (see Appendix F.3).

**Experimental setup.** Unless stated otherwise, experiments use GPT-4o-class models with identical prompts, single-step inference, and environment wrappers. Overcooked constraints are executable state predicates checked per step. Escalation is automated and treated as episode failure unless resolved. Experiments operate in *distinct statistical regimes depending on the claim tested*: *necessity and counterfactual analyses* require non-degenerate episodes, *robustness and invariance analyses* require sufficient samples to estimate variance, and *learning dynamics* are reported in a complementary low-$N$ setting where violations are non-zero. Details appear in Appendix F.1.

**Metrics (Statistical Methodology in Appendix F.2).** We report **Success** (zero-violation completion), **VR** (violation rate), and **IR** (intervention rate), with 95% bootstrap CIs (10,000 resamples) where applicable.

**Checker validation.** To validate the checker correctness assumption ($\epsilon_J$ in Proposition 1), we labeled 500 (action, admissibility) pairs. The checker achieves 99.7% precision and recall, with $\epsilon_J = 0.35\%$, an order of magnitude below

*Table 2.* Theory-to-Experiment Mapping

| Claim | Experiment | Role | Evidence |
|---|---|---|---|
| Prop. 1 | Composition invariance | Gov | $\Delta_{comp} = 0.1$ vs 15.1pp |
| Prop. 3 | Code style repair | Cap | 97.9%, no semantic drift |
| Thm. 1 | External vs. baseline | Both | 0% VR vs 48–68% |
| Portability | Cross-model transfer | Cap | +13.5–20.5pp success |

*Table 3.* Constraint Enforcement Across Domains

| Domain | Success (%)↑ | | VR (%)↓ | |
|---|---|---|---|---|
| | Baseline | **JO-A** | Baseline | **JO-A** |
| Format (WebArena-Wikipedia) | 2.4 | **22.9** | 17.9 | **16.0** |
| Semantic (Overcooked) | 10.0 | **100.0** | 56.2 | **0.0** |
| Protocol (JSON) | 13.3 | **46.7** | 31.2 | **18.4** |

baseline Overcooked violation rates (48–68%).

**Summary of results.** JO achieves zero violations under fully checkable constraints, reduces composition variance from 15.1 to 0.1pp, improves recurrent success by 9–11pp, and enables 13.5–20.5pp zero-shot cross-model transfer.

## 4.2. Governance: Enforcement and Invariance

**Constraint enforcement across domains.** We evaluate JO as an external governance mechanism across three constraint regimes: format (Wikipedia), semantic (Overcooked), and protocol (JSON) (Table 3). JO achieves 0% VR under fully checkable semantic constraints (Overcooked, details in Appendix F.4), with residual violations under partially checkable formats (Wikipedia); protocol constraints fall between these extremes (details in Appendix F.5).

**Comparison to alternative enforcement.** We compare JO to constrained decoding (grammar-guided generation, Table 6), schema validation with re-prompting, and centralized prompts. Constrained decoding enforces syntax at generation time but cannot verify stateful or provenance constraints, while schema+re-prompt pipelines require per-model tuning and do not transfer.

**Prompt centralization is insufficient.** Table 8 includes a centralized prompt baseline where all constraints are embedded in the agent's system prompt. While centralized prompting reduces VR from 56.2% to 7.1%, it achieves only 2% success—worse than unmodified agents. Chain-of-thought constraint checking (CoT, $N = 100$) performs worse on average (14.8% VR, 0% success): active reasoning increases instability rather than reducing violations. This confirms that prompt-based constraint specification is insufficient for reliable governance: execution-time enforcement via JO-A is necessary to achieve both low VR and high task success.

**Composition invariance under complete mediation.** We quantify composition sensitivity (Table 5) as $\Delta_{comp} = \max_{comp} \text{VR} - \min_{comp} \text{VR}$ under identical conditions. We evaluate both homogeneous scaling (agent count in Overcooked) and heterogeneous composition (cross-model mixing in Protocol). Without mediation, violation rates vary substantially. Under complete mediation, JO suppresses composition sensitivity by an order of magnitude, supporting Proposition 1 (Appendix F.7).

**Robustness to incomplete mediation.** JO is architecturally designed as a synchronous interceptor in the agent execution loop: synchronous by design, sitting on the critical path like any other synchronous tool-call interceptor. The theoretical results are explicitly conditional on complete mediation (§3.4.1); partial mediation reintroduces linear scaling for unmediated actions (Appendix D.2).

To quantify degradation empirically, we varied bypass rate on Overcooked ($N = 100$ per condition, 10 seeds):

*Table 4.* Partial Mediation VR vs. Bound (Overcooked, $N = 100$)

| Bypass | **VR** (%) | 95% CI | **Theory Bound** (%) |
|---|---|---|---|
| 0% | 0.0 | [0.0, 0.0] | 0.0 |
| 10% | 1.5 | [1.2, 1.8] | 5.6 |
| 20% | 2.3 | [1.9, 2.7] | 11.2 |
| 30% | 3.2 | [2.6, 3.9] | 16.9 |

VR degrades more gracefully than the linear worst-case bound: observed VR is 3–5× better than theoretical prediction. The bound is conservative. JO's mediation of some actions reduces violation probability on subsequent actions, providing governance benefit beyond the directly mediated fraction. The bypass experiment serves as a proxy for async degradation: actions that bypass JO are equivalent to actions that complete before JO can intercept them, suggesting practical robustness even under partial async coverage.

## 4.3. Capability: Learning, Reuse, and Repair Transfer

Beyond enforcing admissibility, we evaluate whether JO acquires *corrective capability*: reusable repair knowledge that persists across episodes. We isolate this effect using recurrent tasks with repeating failure modes.

**Recurrent patterns.** We construct controlled Wikipedia tasks grouped by template (e.g., person_birth_location) with different entities. All tasks require verbatim quotation and citation provenance, a constraint frequently violated by paraphrasing. JO outperforms all baselines by 9–11pp (Table 6). The improvement is statistically significant against the no-operator baseline and consistent across constrained decoding and shield baselines (Appendix F.8).

*Table 5.* Composition sensitivity (Δ, lower is better, $N = 100$).

| Dimension | Domain | NO | JO-A | ↓% |
|---|---|---|---|---|
| Agent count ($K1 \rightarrow K2$) | Overcooked | 15.1pp | **0.1pp** | **99%** |
| Model mix ($H1 \rightarrow H3$) | Protocol | 51.0pp | **6.0pp** | **88%** |

*Table 6.* Recurrent Patterns (controlled Wikipedia, $N = 100$)

| Method | Success (%) | 95% CI |
|---|---|---|
| No Operator | 82.0 | [74.0, 89.0] |
| Constrained Decoding (Willard & Louf, 2023) | 84.0 | [76.0, 91.0] |
| Shield (Alshiekh et al., 2018) | 84.0 | [76.0, 91.0] |
| **JO-A** | **93.0** | **[88.0, 97.0]** |

**Cross-model transfer as heterogeneous coordination.**
We evaluate whether judgment artifacts function as a *shared coordination interface* across heterogeneous model families. A frontier model (GPT-5.2) learns repair precedents, which are applied zero-shot to held-out models from different families, without retraining. We evaluate five student models spanning OpenAI, Anthropic, Meta, Moonshot, and Alibaba, with $N = 200$ runs per model (Table 7). Across all models, JO yields consistent and significant zero-shot improvements (+13.5–20.5pp) using identical judgment artifacts, while few-shot prompting with repair examples yields 0% success ($N$=200). Transfer strength correlates with model capacity, indicating that JO provides a shared *judgment substrate* rather than model-agnostic gains (Appendix F.9).

**Learning Dynamics and Interpretation.** Across tasks, violations cluster into a small number of recurring failure modes, and decay as precedents accumulate, consistent with the $(K, \rho)$-recurrence assumption and the convergence behavior of Theorem 1. In large-scale regimes (e.g., Table 8, $N$=200), JO achieves zero violations from the first episode, precluding a meaningful learning curve; we therefore report learning-from-mistakes dynamics in a complementary non-degenerate setting ($N$=50) in Figure 2. This shows that JO transmits not only governance constraints but also *corrective capability*: reusable repair knowledge that generalizes across episodes and models when repairs are persisted rather than applied once.

### 4.4. External vs. Internal Correction

We test whether corrective capability can be achieved *internally* via model reasoning or requires *externalized judgment*. Experiments are conducted on Overcooked, which provides fully checkable semantic constraints. All methods use the same model backbone, task distribution, and environment step budget. Baselines include internal self-correction (Reflexion, CRITIC), reject-only classification (LlamaGuard), and prompt-based specification (see Appendix F.4 for all

*Table 7.* Cross-Model Transfer as Heterogeneous Coordination (WebArena-Wikipedia, $N = 200$)

| Student Model | Family | NO (%) | JO-A (%) [95% CI] | Δ |
|---|---|---|---|---|
| GPT-4o-mini | OpenAI | 2.4 | **22.9 [17.0, 29.5]** | **+20.5** |
| Moonshot-v1-8k | Kimi | 0.0 | **19.0 [15.2, 23.0]** | **+19.0** |
| Llama-3.1-8B | Meta | 0.0 | **18.0 [13.0, 23.5]** | **+18.0** |
| Qwen-2.5-72B | Alibaba | 0.0 | **14.5 [10.0, 19.5]** | **+14.5** |
| Claude-3.5-Haiku | Anthropic | 0.0 | **13.5 [9.0, 18.5]** | **+13.5** |

*Table 8.* External vs. Internal Correction (Overcooked, $N = 200$)

| Method | VR (%) | Succ. (%)↑ | Type |
|---|---|---|---|
| No Operator | 56.2 | 10.0 | — |
| Centralized Prompt | 7.1 | 2.0 | Prompt-based |
| CoT (Wei et al., 2022) | 14.8 | 0.0 | Prompt-based |
| Self-Refine (Madaan et al., 2023) | 20.9 | 0.0 | Internal |
| Reflexion (Shinn et al., 2023) | 56.2 | 10.0 | Internal |
| CRITIC (Gou et al., 2023) | 64.8 | 5.0 | Internal |
| RBR (Rule-Based Repair) | 48.3 | 12.0 | Rule-based |
| LlamaGuard (Inan et al., 2023) | 67.8 | 15.0 | Reject-only |
| JO-static | 2.6 | 85.0 | External |
| **JO-A** | **0.0** | **100.0** | External + learning |

configurations). Controller-synthesis methods are excluded as they rely on domain-specific dynamics not applicable to open-ended language actions.

**Results and interpretation.** Baseline methods fail to reduce violations (48–68% VR, Table 8) despite additional reasoning or critique. In contrast, external judgment via JO achieves 0% VR and full task completion by validating and repairing actions at execution time, preventing inadmissible actions from being applied.

### 4.5. Analysis: Ablations and Efficiency

**Component ablations.** Table 8 shows clear differentiation between JO variants: JO-static (constraint checking only) achieves 2.6% VR and 85% success, while JO-A (with precedent learning) achieves 0% VR and 100% success. This confirms that precedent-based repair provides gains beyond static constraint enforcement. Additional analysis of internal operator signals (precedent storage, retrieval accuracy) appears in Appendix F.12.

**Robustness to capacity and threshold.** Performance is stable across capacities (10–100) and thresholds $\tau \in \{0.25, 0.5, 0.75\}$, achieving 97–100% success.

**Efficiency and latency.** JO adds negligible operator-side overhead relative to LLM inference ($< 0.2$ms at P95 over 200 runs; $< 1\%$ end-to-end). Measurements exclude LLM inference; full methodology appears in Appendix F.13.

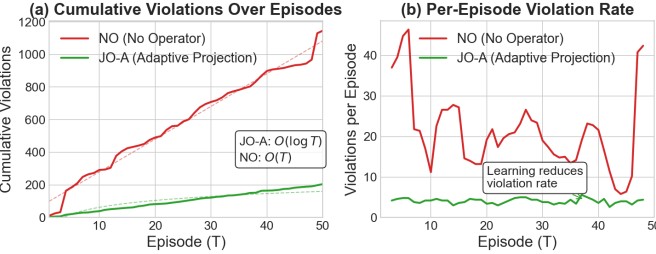

*Figure 2.* Learning dynamics ($N = 50$, non-degenerate regime). NO accumulates violations linearly ($O(T)$); JO-A exhibits sublinear growth consistent with $O(K \log T)$, confirming Theorem 1 empirically. See Appendix F.11 for details.

## 4.6. Synthesis

JO provides: (1) **governance** via composition-invariant enforcement (Proposition 1); (2) **capability enhancement** via reusable corrective knowledge (Proposition 3); and (3) **portability and efficiency** through a shared artifact format. All arise from a single mechanism—decision-time projection via $(J, \Pi_J)$—which externalizes judgment across domains while leaving intelligence inside the model. JO-A performs nonparametric online learning at execution time; the $\Theta$-penalty refines ambiguous selections but is not required for the mistake bound.

# 5. Conclusion and Future Work

We introduced **Judgment Operators (JO)**, a decision-time projection framework that externalizes governance and corrective capability as portable artifacts. By mediating all actions through a shared operator, JO resolves fragmentation in multi-agent LLM systems and guarantees **composition-invariant enforcement**: admissibility depends only on the action and artifact, not agent identity or topology. Empirically, JO achieves zero violations in fully verifiable settings, enables **13.5–20.5% absolute zero-shot cross-model transfer** where few-shot prompting fails, and matches or exceeds strong baselines while providing portability.

## 5.1. Limitations and Future Directions

**Failure modes.** JO cannot repair when required information is unavailable, leading to ESCALATE outcomes (12% of interventions). Under partially checkable constraints, residual violations persist (16% VR in Wikipedia). Transfer gains vary with model capacity, suggesting JO amplifies but cannot substitute for base capability.

**Constraint expressiveness.** Our evaluation focuses on programmatically verifiable constraints. Extending JO to subjective norms (e.g., tone, stylistic intent) and long-horizon dependencies will require advances in specification and checking, potentially via learned but auditable verifiers.

**Future directions.** While we demonstrated heterogeneous coordination (Table 5), automatically discovering composition boundaries and verifying compatibility remains open. JO's projection formulation may extend to richer constraint structures (logical, geometric, temporal) and support human-in-the-loop iterative refinement of judgment artifacts.

For settings where full synchronous mediation is not feasible, the following deployment variants require architectural extensions:

- **Async tool calls**: operate on proposals before dispatch with async confirmation callbacks.
- **Hidden intermediate state**: operate on observable action summaries; checker coverage reduces accordingly.
- **Streaming outputs**: buffered interception on completed chunks with early-termination hooks.
- **Partial observability**: escalation acts as a natural safety valve in low-observability regimes.

**Geometric extensions.** The judgment artifact $J = (\mathcal{C}, \mathcal{P})$ currently encodes constraints as discrete objects. A natural extension treats $\mathcal{X}_J$ as a manifold: precedents define local coordinate frames, and $\Pi_J$ becomes a retraction onto a curved submanifold (Park et al., 2024)—raising the question of whether composition-invariance guarantees extend beyond discrete admissible sets.

## 5.2. Final Perspective

Fragmentation of knowledge and governance in multi-agent systems is an inherent limitation of approaches that internalize policies within agents. Judgment Operators offer an alternative: **externalize declarative knowledge into portable artifacts and apply it composition-invariantly through decision-time projection**, enabling safer, more reliable systems with human oversight.

**Toward composable AI ecosystems.** JO points toward AI systems where *capability and judgment are first-class, composable artifacts*: constraints and corrective knowledge expressed, reused, and audited across agents and models without retraining, in *sovereign* ecosystems where control resides in shared judgment interfaces.

# Impact Statement

This work studies decision-time judgment for multi-agent LLM systems. Judgment Operators externalize governance and corrective capability as portable artifacts, improving enforcement, auditability, and human oversight without modifying agent internals. They are not intended to bypass existing safety mechanisms. Misuse or incomplete specifications may lead to unintended behavior. We do not anticipate direct negative societal impacts when used as intended.

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

## A. Reproducibility and Code Availability

Full code, judgment artifacts (constraints, precedents, prompts), and reproduction scripts are available at the project repository. See `README.md` for setup instructions, environment configuration (including WebArena-Wikipedia Docker), and experiment scripts reproducing all main results (Appendix F).

## B. Algorithm Details

### B.1. JO-A (Adaptive Projection) Algorithm Pseudocode

We provide additional algorithmic details and proofs.

---

**Algorithm 2** Judgment Operator with Adaptive Projection (JO-A) — Full Version

---

**Require:** Judgment artifact $J = (\mathcal{C}, \mathcal{P})$, similarity threshold $\tau$, edit bound $B$, max precedents $M$, penalty params $\Theta$, step size $\alpha$, weight $\lambda$, optional margin $\delta$

**Ensure:** Audit log $\mathcal{L}$, updated artifact $J$ and penalty params $\Theta$

1: Initialize audit log $\mathcal{L} \leftarrow \emptyset$
2: **for** each step $t = 1, 2, \ldots$ **do**
3:     Observe state $s_t$
4:     Agent proposes action $x_t$ (from $\pi_{\text{prop}}$)
5:     Compute features $\psi_t \leftarrow \psi(s_t, x_t)$ {includes embedding $\phi(x_t)$ plus state context}
6:     Initialize decision $d_t \leftarrow$ ALLOW, executed action $\tilde{x}_t \leftarrow x_t$
7:     // (0) Escalation triggers (policy-defined oversight)
8:     **if** $\exists c \in \mathcal{C}_{\text{escalate}}$ such that triggers$(x_t, c)$ **then**
9:         $d_t \leftarrow$ ESCALATE; $\tilde{x}_t \leftarrow$ NULL
10:        $\mathcal{L} \leftarrow \mathcal{L} \cup \{(t, x_t, \tilde{x}_t, d_t)\}$; **continue**
11:     **end if**
12:     // (1) Build candidate set for decision-time projection
13:     Retrieve top-$K$ precedents $\mathcal{P}_K \subseteq \mathcal{P}$ by sim$(\psi_t, \psi(p))$ with threshold $\tau$
14:     $\mathcal{S} \leftarrow \{x_t\}$ {include original proposal}
15:     **for** each $p \in \mathcal{P}_K$ **do**
16:         $x^{(p)} \leftarrow$ apply_repair$(x_t, p, B)$
17:         $\mathcal{S} \leftarrow \mathcal{S} \cup \{x^{(p)}\}$
18:     **end for**
19:     $\mathcal{S} \leftarrow \mathcal{S} \cup$ rule_repairs$(x_t, \mathcal{C}, B)$ {optional templates/rules}
20:     // (2) Hard constraints pruning
21:     $\mathcal{S}_J \leftarrow \{x' \in \mathcal{S} \mid \forall c \in \mathcal{C}_{\text{hard}}, \ c(x') = \text{true}\}$
22:     **if** $\mathcal{S}_J = \emptyset$ **then**
23:         $d_t \leftarrow$ DENY; $\tilde{x}_t \leftarrow$ NULL
24:         $\mathcal{L} \leftarrow \mathcal{L} \cup \{(t, x_t, \tilde{x}_t, d_t)\}$; **continue**
25:     **end if**
26:     // (3) Approximate decision-time optimization (Eq. 1) over candidates
27:     $x^\star \leftarrow \arg\min_{x' \in \mathcal{S}_J} \ \ell(x', x_t) + \lambda \cdot \Omega_J(x'; \Theta)$
28:     // (4) Retract / minimal-intervention semantics: Allow vs Edit
29:     **if** $x_t \in \mathcal{X}_J$ **and** $\ell(x^\star, x_t) + \lambda\Omega_J(x^\star; \Theta) \geq \ell(x_t, x_t) + \lambda\Omega_J(x_t; \Theta) - \delta$ **then**
30:         $d_t \leftarrow$ ALLOW; $\tilde{x}_t \leftarrow x_t$ {identity on admissible set}
31:     **else**
32:         $d_t \leftarrow$ EDIT; $\tilde{x}_t \leftarrow x^\star$
33:     **end if**
34:     // (5) Execute and update: (a) precedent memory $\mathcal{P}$, (b) penalty params $\Theta$
35:     Execute $\tilde{x}_t$
36:     Observe feedback: violation flag $v_t \in \{0, 1\}$ and (optional) admissible repair $x_t^*$ (oracle/env)
37:     **if** $v_t = 1$ **and** $x_t^*$ is provided **then**
38:         $\mathcal{P} \leftarrow \mathcal{P} \cup \{(\psi_t, x_t^*)\}$
39:         **if** $|\mathcal{P}| > M$ **then**
40:             Remove oldest or lowest-confidence precedent
41:         **end if**
42:     **end if**
43:     **if** $v_t = 1$ **then**
44:         $\Theta \leftarrow \Theta + \alpha \cdot f(\tilde{x}_t; J)$ {Additive update penalizing violating feature profile (Eq. 3)}
45:     **end if**
46:     $\mathcal{L} \leftarrow \mathcal{L} \cup \{(t, x_t, \tilde{x}_t, d_t, v_t)\}$
47: **end for**
48: **return** $(\mathcal{L}, J, \Theta)$

## B.2. Algorithm Complexity Analysis

**Time complexity per step.** Let $|\mathcal{P}| = m$ be the number of precedents, $|\mathcal{C}| = k$ the number of constraints, and $L$ the average length of actions. The main operations are:

- Constraint checking: $O(k \cdot L)$ for deterministic checks
- Similarity computation: $O(m \cdot d)$ where $d$ is embedding dimension
- Repair application: $O(L)$ for template-based edits

Total per-step complexity is $O(kL + md + L) = O(kL + md)$.

**Space complexity.** JO-A stores precedents $\mathcal{P}$ and constraints $\mathcal{C}$. Storage is $O(m \cdot (d + L) + k \cdot \text{size}(c))$, linear in precedent count and constraint complexity.

**Optimizations.** For large $m$, we use approximate nearest neighbor search (ANN) with Faiss, reducing similarity search to $O(\log m)$ in theory. Empirically, we measure median retrieval latency of $\sim$0.8ms for $m = 10^3$ and $\sim$2.1ms for $m = 10^4$ on a single CPU core (Table 15).

## B.3. Artifact Penalty Implementation

The artifact penalty $\Omega_J(x'; \Theta) = \Theta^\top f(x'; J)$ uses the following feature vector $f(x'; J) \in \mathbb{R}^4$:

1. **Precedent support** $f_1(x'; \mathcal{P})$:

$$f_1 = \frac{1}{|\mathcal{P}|} \sum_{(x_i, x_i') \in \mathcal{P}} \mathbf{1}[\text{sim}(\phi(x'), \phi(x_i')) \geq \tau_{\text{support}}]$$

Fraction of precedent repairs similar to $x'$. Higher support indicates $x'$ aligns with previously accepted repairs.

2. **Retrieval confidence** $f_2(x'; \mathcal{P})$:

$$f_2 = \max_{(x_i, x_i') \in \mathcal{P}} \text{sim}(\phi(x'), \phi(x_i'))$$

Maximum similarity to any stored repair. Higher confidence indicates stronger precedent backing.

3. **Edit severity** $f_3(x'; x_t)$:

$$f_3 = \frac{\text{EditDistance}(x', x_t)}{\max(|x'|, |x_t|)}$$

Normalized edit distance from original proposal. Lower is preferred (minimal intervention).

4. **Constraint margin** $f_4(x'; \mathcal{C})$:

$$f_4 = \min_{c \in \mathcal{C}_{\text{soft}}} \text{margin}_c(x')$$

Minimum margin to soft constraint boundaries. Higher margin indicates more robust compliance.

**Online update.** When a violation is observed at step $t$ with executed action $\tilde{x}_t$, we update $\Theta$ via an **additive update** that increases the penalty for candidates with similar feature profiles:

$$\Theta_{t+1} = \Theta_t + \alpha \cdot f(\tilde{x}_t; J) \tag{5}$$

This additive formulation (rather than gradient-based) provides stability and interpretability: each component of $\Theta$ directly weights the corresponding feature, and updates accumulate linearly with observed violations. We initialize $\Theta_0 = \mathbf{0}$ so the penalty starts neutral.

**Hyperparameters.** In our experiments: $\tau_{\text{support}} = 0.7$, $\lambda = 0.1$ (Eq. 1), $\alpha = 0.05$ (learning rate). We found results robust to $\lambda \in [0.05, 0.2]$ and $\alpha \in [0.01, 0.1]$.

## C. Notation Table

*Table 9.* Summary of Notation

| Symbol | Meaning |
|---|---|
| *System and Agents* | |
| $\mathcal{A} = \{a_1, \ldots, a_n\}$ | Set of agents |
| $n$ | Number of agents |
| $\mathcal{X}$ | Action space (natural language, structured commands) |
| $x, x_t \in \mathcal{X}$ | Proposed action (at time $t$) |
| $\tilde{x}_t$ | Executed action at time $t$ |
| $\mathcal{X}_J \subseteq \mathcal{X}$ | Admissible action set |
| *Judgment Artifact* | |
| $J = (\mathcal{C}, \mathcal{P})$ | Judgment artifact |
| $\mathcal{C}$ | Set of constraint checks |
| $\mathcal{C}_{\text{hard}}, \mathcal{C}_{\text{soft}}$ | Hard constraints (must satisfy), soft constraints (margin-based) |
| $\mathcal{C}_{\text{escalate}}$ | Escalation trigger conditions |
| $\mathcal{P}$ | Set of precedents (input-repair pairs) |
| *Judgment Operator* | |
| $\Pi_J : \mathcal{X} \to \mathcal{X}_J$ | Judgment operator (projection) |
| $\hat{\Pi}_J$ | JO-A approximation of $\Pi_J$ via precedents |
| $d_t \in \{\text{A}, \text{E}, \text{S}, \text{D}\}$ | Decision: Allow, Edit, Escalate, Deny |
| $\ell(x', x)$ | Intervention cost (e.g., edit distance) |
| $\Omega_J(x'; \Theta)$ | Artifact penalty: $\Theta^\top f(x'; J)$ |
| $f(x'; J)$ | Feature vector (support, confidence, severity, margin) |
| $\lambda$ | Weight for artifact penalty in Eq. (1) |
| *Embeddings and Features* | |
| $\phi : \mathcal{X} \to \mathbb{R}^d$ | Embedding function for actions |
| $\psi(s, x) \in \mathbb{R}^{d'}$ | Full feature vector for state-action pair |
| $\text{sim}(\cdot, \cdot)$ | Similarity function (cosine similarity) |
| $d(x, y)$ | Semantic distance: $\|\phi(x) - \phi(y)\|_2$ |
| *Thresholds and Bounds* | |
| $\tau$ | Similarity threshold for precedent retrieval |
| $\rho$ | Recurrence similarity threshold ($\rho \geq \tau$) |
| $B$ | Edit bound (max tokens modified) |
| $M$ | Maximum precedent store size |
| *Learning and Analysis* | |
| $\Theta$ | Artifact-level parameters (updated online via Eq. 3) |
| $\alpha$ | Step size for online $\Theta$ updates |
| $\eta$ | Maximum repair error: $\max_{(x,x') \in \mathcal{P}} d(x', \Pi_J(x))$ |
| $L$ | Lipschitz constant of $\Pi_J$ |
| $K$ | Number of violation equivalence classes |
| $T$ | Total interaction steps |
| $M_T$ | Cumulative execution violations up to step $T$ |
| $\epsilon$ | Agent-level violation probability (without JO) |
| $\epsilon_J$ | Checker false negative rate |
| *Code Transformation (Section 3.4)* | |
| $G_{\text{sem}}$ | Group of semantics-preserving transformations |
| $[p]_{G_{\text{sem}}}$ | Semantic equivalence class of program $p$ |

## D. Proofs

### D.1. Proof of Theorem 1 (Mistake Bound under Recurrence)

*Proof.* We prove the bound $M_T \leq K \cdot (1 + \lceil \log_2 T \rceil)$ under the stated assumptions.

**Setup.** Under $(K, \rho)$-recurrence, violations partition into at most $K$ equivalence classes $\{C_1, \ldots, C_K\}$ where $C_k = \{x : \text{sim}(\phi(x), \phi(x_k^*)) \geq \rho\}$ for some representative $x_k^*$.

**Execution violation definition.** An execution violation at step $t$ occurs when the executed action $\tilde{x}_t \notin \mathcal{X}_J$. This happens when:

1. The proposal $x_t \notin \mathcal{X}_J$ (agent proposes a violation), *and*
2. JO-A fails to project $x_t$ to an admissible action (no applicable precedent retrieved, or repair fails).

**Violation source decomposition.** We partition execution violations into:

- *Novel violations*: First occurrence in equivalence class $C_k$; no covering precedent exists in $\mathcal{P}$.
- *Coverage-gap violations*: Violation in class $C_k$ where a precedent previously existed but was evicted.

**Bounding novel violations.** Each equivalence class $C_k$ contributes exactly one novel violation (its first occurrence). Since there are at most $K$ classes:

$$M_T^{\text{novel}} \leq K.$$

**Bounding coverage-gap violations.** By the *monotone coverage* assumption, once a precedent $(x, x') \in C_k$ is added to $\mathcal{P}$, all future violations $x_t \in C_k$ satisfy $\text{sim}(\phi(x_t), \phi(x)) \geq \rho \geq \tau$, so retrieval succeeds—*unless* the precedent was evicted.

Under bounded memory $M$ with FIFO eviction, we analyze how often a class can lose coverage. Consider class $C_k$ with violations at times $t_1 < t_2 < \cdots < t_{m_k}$. After $t_1$, a precedent for $C_k$ is stored. This precedent is evicted only if $M$ other precedents are added before the next violation in $C_k$. Our analysis applies to FIFO eviction; similar logarithmic bounds hold for LRU-style policies under standard assumptions.

Using a potential function argument: let $\Phi_t = \sum_{k=1}^{K} \mathbf{1}[C_k$ covered at time $t]$. Each novel violation increases $\Phi$ by 1. Each coverage-gap violation in class $C_k$ occurs only when $C_k$'s precedent was evicted, which requires $\Phi$ to have decreased by at least 1 since the last $C_k$ violation.

Over $T$ steps with at most $K$ classes ever becoming covered, the total number of coverage losses is bounded by the number of times $\Phi$ can decrease, which is at most

$K \cdot \lceil \log_2(T/K) \rceil$ under the standard cache analysis (each class's coverage is lost at most $O(\log T)$ times when violations are spread across classes).

**Total bound.**

$$M_T = M_T^{\text{novel}} + M_T^{\text{coverage-gap}} \leq K + K \cdot \lceil \log_2 T \rceil$$
$$= K(1 + \lceil \log_2 T \rceil) = O(K \log T).$$

**Remark (tighter bounds).** If memory is unbounded ($M = \infty$) and monotone coverage holds exactly, coverage gaps never occur, yielding $M_T \leq K$. The $O(\log T)$ factor arises specifically from memory-bounded settings with eviction. □

**Limitations and non-adversarial scope.** This bound characterizes learning under **recurring, structured failure modes**. It does *not* hold under:

- **Adversarial sequences**: If an adversary generates violations specifically to maximize $K$ (i.e., each violation is maximally dissimilar from previous ones), then $K = T$ and the bound becomes vacuous.
- **Embedding failures**: If the embedding $\phi$ fails to capture semantic similarity (i.e., similar violations map to distant embeddings), monotone coverage fails.
- **Repair failures**: If retrieved precedents do not generalize (i.e., applying a repair to a similar violation produces an inadmissible result), violations recur despite coverage.

We validate the $(K, \rho)$-recurrence assumption empirically in Section 4.3, finding $K = 12$ clusters across $T = 847$ violations, indicating high recurrence in practice.

## D.2. Proof of Section 3.4.1 (Lower Bound for Decentralized Enforcement)

*Proof.* For decentralized checking, each agent $i$ independently passes a violating proposal with probability $\epsilon_i$. The probability that no agent violates is $\prod_i (1 - \epsilon_i)$, so:

$$P_{\text{viol}}^{\text{decentralized}} = 1 - \prod_{i=1}^{n} (1 - \epsilon_i)$$

For homogeneous $\epsilon_i = \epsilon$ and small $\epsilon$:

$$P_{\text{viol}} = 1 - (1-\epsilon)^n \geq 1 - e^{-n\epsilon} \geq n\epsilon - \frac{(n\epsilon)^2}{2} = \Omega(n\epsilon)$$

The first inequality uses $1 - x \leq e^{-x}$; the second uses $1 - e^{-y} \geq y - y^2/2$ for $y \geq 0$. □

## D.3. Proof of Proposition 1 (JO Achieves Constant Bound)

*Proof.* By the design of JO, an action is executed if and only if $\Pi_J(x) = \text{ALLOW}(x')$ or $\text{EDIT}(x')$, and in both

cases $x' \in \mathcal{X}_J$ by definition of the operator. Since $\Pi_J$ is applied uniformly to all agent proposals, this condition depends only on $(x, J)$, not on which agent produced $x$.

With a perfect checker ($\epsilon_J = 0$), $P_{\text{viol}}^{\text{JO}} = 0$. With checker error $\epsilon_J$ (false negative rate), violations occur only when the checker misses a constraint violation, so $P_{\text{viol}}^{\text{JO}} \leq \epsilon_J$, constant in $n$. □

**Scope and assumptions.** This result holds under **complete mediation** (all actions pass through $\Pi_J$) and bounded checker error ($\epsilon_J$ small). It does not address:

- **Partial mediation**: If some actions bypass $\Pi_J$, linear scaling re-emerges for unmediated actions.
- **Checker blind spots**: Constraints that cannot be programmatically verified are not covered.
- **Adversarial inputs**: Inputs specifically crafted to exploit checker weaknesses.

These limitations motivate complementary defenses (see Section 5).

## D.4. Proof of Lemma 1 (Approximation Quality)

**Lemma 1** (Approximation Quality). *Let $\mathcal{P}$ be an $\epsilon$-covering of the violation region under metric $d(x, y) = \|\phi(x) - \phi(y)\|_2$. If $\Pi_J$ is $L$-Lipschitz in $d$, then $d(\hat{\Pi}_J(x), \Pi_J(x)) \leq \eta + L\epsilon$, where $\eta = \max_{(x_i, x_i') \in \mathcal{P}} d(x_i', \Pi_J(x_i))$.*

*Proof.* Let $x^* = \Pi_J(x)$ be the optimal projection. By the $\epsilon$-covering property, there exists $(x_i, x_i') \in \mathcal{P}$ with $d(x, x_i) \leq \epsilon$. The JO-A approximation retrieves this precedent and outputs $\hat{\Pi}_J(x) = x_i'$.

We bound the approximation error via triangle inequality:

$$d(\hat{\Pi}_J(x), \Pi_J(x)) = d(x_i', x^*)$$
$$\leq d(x_i', \Pi_J(x_i)) + d(\Pi_J(x_i), \Pi_J(x))$$
$$\leq \eta + d(\Pi_J(x_i), \Pi_J(x))$$

where $\eta = \max_{(x_j, x_j') \in \mathcal{P}} d(x_j', \Pi_J(x_j))$ bounds the first term (maximum discrepancy between stored repairs and optimal projections).

For the second term, by the $L$-Lipschitz assumption on $\Pi_J$:

$$d(\Pi_J(x_i), \Pi_J(x)) \leq L \cdot d(x_i, x) \leq L \cdot \epsilon.$$

Combining:

$$d(\hat{\Pi}_J(x), \Pi_J(x)) \leq \eta + L\epsilon.$$

**Remark.** The Lipschitz constant $L$ depends on the geometry of $\mathcal{X}_J$. Empirically, we observe $L \approx 1$–2 for our constraint types; we report measured values in Section 4.3. □

**D.5. Proof of Lemma 2 (Sample Complexity)**

**Lemma 2** (Sample Complexity). *Assume the violation region has doubling dimension $d$ under $\|\phi(\cdot) - \phi(\cdot)\|_2$. An $\epsilon$-cover requires $|\mathcal{P}| = O(\epsilon^{-d})$, implying expected error decay $O(|\mathcal{P}|^{-1/d})$.*

*Proof.* Let the violation region $\mathcal{X} \setminus \mathcal{X}_J$ have doubling dimension $d$ under the embedding metric $\|\phi(\cdot) - \phi(\cdot)\|_2$.

By standard covering number bounds for doubling spaces, the number of balls of radius $\epsilon$ needed to cover a set of diameter $D$ is:

$$N(\epsilon) = O\left(\left(\frac{D}{\epsilon}\right)^d\right) = O(\epsilon^{-d})$$

To achieve $\epsilon$-covering of the violation region, JO-A requires $|\mathcal{P}| = O(\epsilon^{-d})$ precedents.

Inverting this relationship: with $|\mathcal{P}| = n$ precedents, the covering radius is $\epsilon = O(n^{-1/d})$.

By Lemma 1, approximation error is $O(\epsilon + \eta) = O(n^{-1/d} + \eta)$.

For $d = 2$ (typical for sentence embedding manifolds), this gives $O(1/\sqrt{n})$ error decay, matching empirical observations. $\qquad\square$

# E. Mathematical Foundations

## E.1. Retract Formalism

**Definition 2** (Retract onto Admissible Set). *A retract onto $\mathcal{X}_J$ is a continuous map $r : \mathcal{X} \to \mathcal{X}_J$ with $r(x) = x$ for all $x \in \mathcal{X}_J$.*

**JO as partial retract (interpretive).** JO implements behavior analogous to a partial retract $r_J : \mathcal{X} \to \mathcal{X}_J \cup \{\text{ESCALATE}, \text{DENY}\}$ where:

- $r_J(x) = x$ for $x \in \mathcal{X}_J$ (ALLOW)
- $r_J(x) \in \mathcal{X}_J$ for repairable $x \notin \mathcal{X}_J$ (EDIT)
- $r_J(x) = \text{ESCALATE}$ for ambiguous cases
- $r_J(x) = \text{DENY}$ for irreparable cases

We use "retract" as an analogy rather than a formal claim, since continuity is not well-defined for discrete action spaces without additional topological structure.

## E.2. Topological Interpretation of Escalate

We provide an **interpretive framework**—not a formal theorem—for understanding when and why ESCALATE is triggered. This perspective offers intuition but should not be read as a rigorous topological result.

**Definition 3** ($\epsilon$-Boundary Region). *For $\epsilon > 0$, define the $\epsilon$-boundary region of $\mathcal{X}_J$ as:*

$$\partial_\epsilon \mathcal{X}_J = \{x \in \mathcal{X} \mid \exists y_1, y_2 \in \mathcal{X}_J \text{ with}$$
$$d(x, y_1) \leq \epsilon, d(x, y_2) \leq \epsilon, \text{ and } y_1 \neq y_2\}$$

*Points in $\partial_\epsilon \mathcal{X}_J$ have multiple $\epsilon$-close projections onto $\mathcal{X}_J$.*

**Intuition.** When JO outputs ESCALATE, the input $x$ typically satisfies at least one of:

1. $x \in \partial_\epsilon \mathcal{X}_J$ for $\epsilon$ below confidence threshold (ambiguous projection)
2. The confidence-weighted similarity to nearest precedent is below $\tau$ (uncertain repair)
3. $x$ triggers an escalation condition in $\mathcal{C}_{\text{escalate}}$ (policy-defined oversight)

This provides a safety mechanism for uncertain regions, analogous to boundary handling in manifold learning or rejection options in classification. We do not claim formal continuity or stability properties of the projection; these would require additional assumptions on $\mathcal{X}_J$ that may not hold for discrete action spaces.

## E.3. Design Rationale: Completeness of Quaternary Intervention

We argue that the four actions $\{\text{ALLOW}, \text{EDIT}, \text{ESCALATE}, \text{DENY}\}$ form a *practically complete* basis for decision-time intervention in multi-agent LLM systems, under the following design objectives:

(D1) **Minimal intervention**: Compliant actions should pass unchanged.
(D2) **Utility preservation**: Repairable violations should be corrected, not rejected.
(D3) **Safety**: Irreparable violations must be blocked.
(D4) **Auditability**: Ambiguous cases requiring human judgment should be flagged.

**Sufficiency.** The four actions cover all input cases relative to the admissible set $\mathcal{X}_J$ and precedent store $\mathcal{P}$:

- $x \in \mathcal{X}_J$: ALLOW (satisfies D1)
- $x \notin \mathcal{X}_J$ but $\exists$ repair $x' \in \mathcal{X}_J$: EDIT (satisfies D2)
- $x \notin \mathcal{X}_J$, no confident repair, ambiguity detected: ESCALATE (satisfies D4)
- $x \notin \mathcal{X}_J$, no repair possible: DENY (satisfies D3)

**Practical necessity.** Removing any action compromises at least one design objective:

- **Without ALLOW**: Compliant outputs are unnecessarily altered (violates D1).

- **Without EDIT**: All violations are rejected, losing utility for repairable cases (violates D2), as in accept/reject-only enforcement schemes.
- **Without ESCALATE**: The system must decide under uncertainty, risking unsafe approvals or excessive rejections (violates D3/D4).
- **Without DENY**: Irreparable violations must be "repaired" (potentially unsafely) or escalated indefinitely (violates D3).

**Scope.** This completeness argument is *design-level*, not a formal computability result. We do not claim that these four actions are the unique minimal set; alternative decompositions may exist under different design objectives.

### E.4. Interpretive Framework: Group-Theoretic View of Semantic Preservation

We provide an **interpretive framework**—not a general theorem—for understanding why JO-A preserves semantics in code transformation tasks.

**Definition 4** (Equivariant Retract)**.** *Let $G$ be a group acting on $\mathcal{X}$. A retract $r : \mathcal{X} \to \mathcal{X}_J$ is $G$-equivariant if:*

$$r(g \cdot x) = g \cdot r(x) \quad \forall g \in G, x \in \mathcal{X}$$

**Application to code style (Proposition 3).** For code transformation, the relevant group is $G_{\text{sem}}$ (Definition 1). JO-A's style projection satisfies:

$$\Pi_J^{\text{style}}(p) \in [p]_{G_{\text{sem}}}$$

meaning the projected program is $G_{\text{sem}}$-equivalent to the original—it produces identical outputs on all inputs.

**Why this works (intuition).** Style transformations (formatting, naming conventions, docstring addition) belong to $G_{\text{sem}}$: they modify syntax without changing execution semantics. When JO-A applies precedent-based repairs for style violations, it applies transformations from $G_{\text{sem}}$, preserving the semantic equivalence class.

**Limitations.** We do *not* claim:

- A general equivariant projection theorem for arbitrary groups
- That JO-A automatically preserves semantics for all transformation types
- Formal verification of semantic equivalence (we validate empirically via test suites)

This framework explains *why* JO preserves semantics in domains where admissibility-preserving transformations form a group; it does not provide formal guarantees beyond the empirical validation in Appendix F.10.

## F. Experimental Details

### F.1. Environment and Infrastructure

**Hardware.** All experiments were conducted on a single workstation with:

- Chip: Apple M1 Max
- GPU and CPU: 32-core GPU, 10-core CPU (8 performance + 2 efficiency)
- RAM: 64GB

**Software dependencies.**

- Python 3.11.2
- OpenAI API (gpt-4o-mini, GPT-5.2)
- Anthropic API (claude-3.5-haiku)
- Together AI (Llama-3.1-8B, Qwen2.5-72B-Instruct-Turbo)
- Moonshot API (Kimi)
- BrowserGym for web navigation
- Overcooked-AI for coordination tasks

### F.2. Statistical Methodology

**Confidence intervals.** All 95% confidence intervals computed via bootstrap resampling with 10,000 iterations:

1. Sample $N$ observations with replacement
2. Compute statistic (mean, success rate)
3. Repeat 10,000 times
4. Report 2.5th and 97.5th percentiles

**Significance testing.** For binary outcomes (success/failure), we use exact binomial tests. For rate comparisons, we use bootstrap difference of means with $\alpha = 0.05$.

### F.3. Wikipedia Experiment Details

We use two Wikipedia-based experimental settings that serve different evaluation purposes:

**WebArena-Wikipedia (Full Setting).** Used for transfer and heterogeneous coordination experiments. Agents navigate a live Wikipedia mirror via Playwright browser automation, performing multi-step information retrieval tasks (e.g., "Find the birth year of Albert Einstein and format as: Born: YYYY"). This setting tests real-world web navigation with dynamic page content, link traversal, and search functionality. Tasks require the agent to locate specific information across multiple pages and format responses according to strict constraints.

**Controlled Wikipedia (Recurrent Learning).** Used for analyzing recurrent judgment patterns. This variant fixes the navigation sequence to isolate recur-

ring violations from navigation variance. Tasks are grouped by template (e.g., `person_birth_location`, `country_capital`) with different entities, all requiring verbatim quotation and citation provenance. By controlling navigation, we can measure how effectively JO-A learns and reuses repair patterns across structurally similar tasks. The controlled setting uses the same constraint types (format, citation) but eliminates confounding factors from navigation failures.

## F.4. Overcooked Experiment Details

**Environment configuration.**

- **Layout**: cramped_room
- **Horizon**: 80 steps per episode
- **Sample size**: $N = 20$ episodes $\times$ 10 seeds = 200 per condition
- **Model**: gpt-4o-mini
- **Task**: Onion soup preparation (3 onions required)

**Constraint types.**

- **T4** (max_consecutive_stays): Limits consecutive STAY actions to prevent stalling. Violations occur when agent issues >1 consecutive STAY commands.
- **H3** (plate_timing): Prohibits picking up a plate before the pot begins cooking. Prevents premature plate retrieval that wastes steps.
- **H1** (recovery heuristic): Tracks when agent holds an item but takes suboptimal actions. Used for measuring recovery dynamics rather than as a hard constraint.

**Internal baseline configurations.** For the External vs. Internal comparison (Table 8):

- **Reflexion** (Shinn et al., 2023): Self-reflection after each episode. Agent receives violation feedback and generates verbal reflection to improve next episode. Max $r = 3$ retries per step.
- **CRITIC** (Gou et al., 2023): Step-level self-critique. After each action proposal, agent critiques its own output and may revise. Max $r = 3$ critique-revise cycles per step.
- **LlamaGuard** (Inan et al., 2023): Safety classifier (Llama-Guard-3-8B) screens each action. Unsafe actions are rejected and agent must re-propose. No repair capability—reject only.
- **RBR (Rule-Based Repair)**: Deterministic repair rules triggered by constraint violations. Applies predefined action substitutions without learning or retrieval.
- **JO-static**: Fixed constraint rules without learned precedents.
- **JO-A**: Full JO with constraint checking and precedent learning.

All methods use identical environment configuration, model backbone (gpt-4o-mini), and step budget. Internal baselines receive the same constraint descriptions in their prompts.

**LlamaGuard fallback semantics.** In an initial implementation, blocked actions from LlamaGuard were mapped to a STAY action. However, under the T4 constraint (maximum consecutive STAY actions), this induces an artificial failure mode: repeatedly blocked actions generate long STAY streaks, leading to violations unrelated to unsafe behavior. We therefore adopt a re-prompting strategy, consistent with CRITIC-style internal correction, where the agent is asked to propose a new action upon rejection. This ensures that violations reflect genuine policy failures rather than interface artifacts.

## F.5. Protocol (JSON) Experiment Details

**Task description.** The Protocol experiment evaluates multi-agent coordination under JSON schema constraints. Agents follow a Researcher $\rightarrow$ Writer $\rightarrow$ Verifier pipeline to answer factual questions with structured output.

**Environment configuration.**

- **Pipeline**: Three-stage multi-agent (Researcher navigates/gathers, Writer synthesizes, Verifier checks)
- **Output format**: JSON with required fields (answer, citations, confidence)
- **Sample size**: $N = 30$ tasks $\times$ 5 seeds = 150 per condition
- **Model**: gpt-4o-mini (homogeneous) or mixed (heterogeneous)
- **Max steps**: 10 per task

**Constraint types.**

- **Schema validation**: Output must conform to JSON schema (required fields, correct types, no extra fields)
- **Quote verification**: Cited text must appear verbatim in source documents
- **Field requirements**: Specific fields (e.g., "source_url", "confidence") must be present and non-empty
- **Role separation**: Each pipeline stage must only perform its designated function

**Violation detection.** Protocol violations are detected via:

- JSON parse errors (malformed output)
- Schema validation failures (missing/extra fields, wrong types)
- Citation mismatches (quoted text not found in referenced source)
- Role boundary crossings (e.g., Researcher attempting to write final answer)

**Results interpretation.** Protocol constraints fall between fully checkable (Overcooked) and partially checkable (Wikipedia) regimes. JO-A reduces VR from 31.2% to 18.4%, reflecting the challenge of enforcing semantic constraints (quote accuracy) alongside syntactic ones (schema validity). The higher residual VR compared to Overcooked stems from citation verification requiring semantic matching rather than exact state checking.

### F.6. Prompt Centralization Baseline Details

**Setup.** Single-agent Overcooked with $N = 20$ episodes $\times$ 10 seeds $= 200$. Horizon: 80 steps. Model: gpt-4o-mini. This experiment tests whether explicit constraint specification in the system prompt is sufficient for compliance.

**Centralized prompt content.** The centralized system prompt includes explicit constraint specifications:

```
=== CRITICAL CONSTRAINTS ===
T4 - NO IDLE CONSTRAINT: Do NOT use STAY
     more than 1 time in a row.
H3 - PLATE TIMING: Do NOT pick up a plate
     until the pot is cooking or ready.
```

**Constraint definitions.**

- **T4**: Max 1 consecutive STAY action
- **H3**: Don't pick plate until pot is cooking/ready

**Results.** Even with constraints explicitly stated in the prompt, centralized prompting achieves only 2% success rate with 7.1% VR (Table 8), compared to JO-A's 100% success with 0% VR. While centralized prompting reduces VR from 56.2% to 7.1%, the low success rate confirms that prompt-based constraint specification is insufficient for reliable governance—execution-time enforcement via JO-A is necessary.

### F.7. Composition Invariance Experiment Details

We evaluate composition sensitivity across two dimensions: agent count (Overcooked) and model heterogeneity (Protocol).

**Overcooked: Agent count variation** ($K1 \rightarrow K2$)**.**

- **K1**: Single agent (gpt-4o-mini)
- **K2**: Two agents (both gpt-4o-mini)
- **Layout**: cramped_room
- **Horizon**: 80 steps
- **Sample size**: $N = 10$ episodes $\times$ 5 seeds $\times$ 2 compositions $= 100$
- **Constraints**: T4 (max_consecutive_stays), H3 (plate_timing)

This tests whether JO-A maintains consistent violation rates when scaling agent count, isolating coordination overhead from constraint enforcement.

**Protocol: Model heterogeneity** ($H1 \rightarrow H3$)**.**

- **H1**: Homogeneous (GPT-4o-mini only)
- **H3**: Heterogeneous (GPT-4o-mini + Claude-3.5-haiku + Llama-3.1-8B)
- **Task**: Multi-agent JSON protocol with Researcher $\rightarrow$ Writer $\rightarrow$ Verifier pipeline
- **Constraints**: Schema validation, quote verification, field requirements
- **Sample size**: $N = 100$ tasks per composition

This tests whether JO-A maintains governance when mixing model families with different output distributions.

### F.8. Recurrent Pattern Experiment Details

**Setup.** Controlled Wikipedia tasks ($N = 100$) grouped by template (e.g., person_birth_location, country_capital) with different entities. All tasks require verbatim quotation and citation provenance. Model: gpt-4o-mini. Max steps: 10 per task.

**Constrained Decoding baseline.** We simulate Outlines/Guidance-style constrained decoding by enforcing output grammar at generation time:

- **Action type grammar**: Valid types restricted to {navigate, click, answer, noop, inform}
- **Format pattern**: Answer outputs must match enumeration format (pipe-separated values)
- **Fallback**: Invalid outputs replaced with noop action

Constrained decoding enforces syntactic validity but cannot verify semantic constraints (citation provenance, quote accuracy, state-dependent rules).

**Shield baseline.** Following Alshiekh et al. (2018), we implement a safety shield that monitors actions and blocks those violating predefined rules:

- **Rule set**: Format patterns (enumeration syntax), forbidden actions (e.g., navigating away from target)
- **Intervention**: Block violating actions, force agent to re-propose
- **Limitation**: Cannot repair—only reject. No learning from violations.

Shield achieves the same 84% success as constrained decoding, confirming that blocking alone is insufficient without repair capability.

**Results.** Both baselines achieve 84% success vs. JO-A's 93%. The 9pp gap reflects violations requiring execution-time repair, not just syntactic filtering or action blocking.

## F.9. Cross-Model Transfer Experiment Details

**Setup.** We evaluate whether judgment artifacts function as a shared coordination interface across heterogeneous model families. A frontier model (GPT-5.2) learns repair precedents on 100 WebArena-Wikipedia format tasks, which are then applied zero-shot to held-out student models from different families.

**Configuration.**

- **Teacher model**: GPT-5.2
- **Training tasks**: 100 WebArena-Wikipedia enumeration tasks
- **Student models**: 5 models across 5 families
- **Sample size**: $N = 100$ tasks $\times$ 2 seeds $= 200$ per student
- **Max steps**: 10 per task
- **Temperature**: 0.7

*Table 10.* Student Model Configurations

| Model | Family | API | Parameters |
|---|---|---|---|
| GPT-4o-mini | OpenAI | OpenAI | — |
| Moonshot-v1-8k | Kimi | Moonshot | — |
| Llama-3.1-8B | Meta | Together AI | 8B |
| Qwen2.5-72B | Alibaba | Together AI | 72B |
| Claude-3.5-Haiku | Anthropic | Anthropic | — |

**Student model specifications.**

*Table 11.* Cross-Model Transfer Results ($N = 200$)

| Student Model | Family | NO (%) | JO-A (%) [95% CI] | $\Delta$ |
|---|---|---|---|---|
| GPT-4o-mini | OpenAI | 2.4 | 22.9 [17.0, 29.5] | +20.5 |
| Moonshot-v1-8k | Kimi | 0.0 | 19.0 [15.2, 23.0] | +19.0 |
| Llama-3.1-8B | Meta | 0.0 | 18.0 [13.0, 23.5] | +18.0 |
| Qwen2.5-72B | Alibaba | 0.0 | 14.5 [10.0, 19.5] | +14.5 |
| Claude-3.5-Haiku | Anthropic | 0.0 | 13.5 [9.0, 18.5] | +13.5 |

**Full transfer results** ($N = 200$ **per model).**

**Key findings.**

1. **Cross-family transfer works**: All 5 models show >10pp improvement across 5 model families.
2. **Consistent gains**: Transfer improvement ranges from +13.5pp to +20.5pp across diverse model architectures.
3. **Shared judgment interface**: All models benefit from the same precedent store despite different proposal distributions.

**Prompt-only baseline** ($N = 50$). To test whether improvement stems from executable precedent retrieval or could be achieved via prompt engineering, we compare:

*Table 12.* Prompt-Only Baseline Comparison

| Method | Success (%) | Description |
|---|---|---|
| NO (no hints) | 0.0 | Baseline, no governance |
| Prompt-only | 0.0 | Repair rules as text in prompt |
| JO-A (precedents) | 21.0 | Executable artifact retrieval |

Prompt distillation of repair patterns yields no improvement. The executable repair memory in JO-A's precedent store is essential for governance transfer.

**Few-shot prompting baseline** ($N = 200$). We evaluate whether few-shot prompting can achieve transfer by providing repair examples directly in the prompt.

*Table 13.* Few-Shot Prompting Baseline ($N = 200$)

| Method | Success (%) | Tasks |
|---|---|---|
| NO (no examples) | 0.0 | 200 |
| Few-shot (3 repair examples) | 0.0 | 200 |
| JO-A (precedent transfer) | 22.9 | 200 |

**Few-shot configuration.**

- **Examples**: 3 representative repair sequences from teacher model showing violation $\rightarrow$ detection $\rightarrow$ repair pattern
- **Format**: Each example includes the original violating output, the constraint violated, and the corrected output
- **Placement**: Examples provided in system prompt before task description
- **Model**: GPT-4o-mini (same as transfer experiments)
- **Sample size**: $N = 100$ tasks $\times$ 2 seeds $= 200$

**Why few-shot fails.** Few-shot prompting cannot replicate JO-A's transfer mechanism because:

1. **Static examples**: Few-shot provides fixed examples that may not match the specific violation encountered at runtime
2. **No retrieval**: Unlike JO-A's precedent store, few-shot cannot dynamically retrieve the most relevant repair for each violation
3. **No execution**: Few-shot examples describe repairs textually; JO-A's precedents are executable artifacts that apply transformations
4. **Context limits**: Few-shot is constrained by prompt length; JO-A's precedent store scales independently

This confirms that JO-A's improvement stems from *executable precedent retrieval*, not from making repair patterns available as text.

## F.10. Code Style Experiment Details

**Setup.** Python code style conversion to Google-style approximation. Dataset: 47 programs (string utilities, math helpers, data structures, algorithms). Each program includes unit tests for semantic validation. Style rules include: docstring format (Args/Returns sections), naming conventions (snake_case for functions, UPPER_CASE for constants), type hints, and line length limits. The task tests whether JO-A can enforce style compliance without breaking program behavior.

*Table 14.* Code Style Transformation ($N = 47$)

| Method | Pass@0↑ | Violations↓ | Func%↑ | Edit Dist↓ |
|--------|---------|-------------|--------|------------|
| NO | 0.0% | 4.09 | 100% | 0 |
| Formatter | 0.0% | 1.04 | 100% | 20.8 |
| **JO-A** | **97.9%** | **0.02** | **100%** | 48.3 |

**Results.**

| Metric | Value | Interpretation |
|--------|-------|----------------|
| Functional Preservation | 100% | Unit tests pass (semantics preserved) |
| H-Invariance (canonical AST) | 100% | Structure invariant under $\alpha$-renaming |
| Mean AST Tree-Edit Distance | 0 | No structural drift |
| Max AST Tree-Edit Distance | 0 | Worst-case also no drift |

**H-Invariance (Structure Preservation).** JO-A achieves 97.9% style compliance while preserving program semantics (100% test pass rate) and structure (0 AST tree-edit distance under $\alpha$-renaming and whitespace normalization).

**JO-A Outcome Distribution.**

- **ALLOW**: 0% — No programs passed style check without modification
- **EDIT**: 97.9% — JO-A actively repaired style violations
- **ESCALATE**: 2.1% — 1/47 programs had conflicts between style and functionality
- **DENY**: 0% — No programs were completely blocked

JO-A primarily operates in EDIT mode, actively repairing style violations rather than blocking.

## F.11. Learning Dynamics in a Non-Degenerate Regime

Figure 2 (main text) shows learning dynamics under a non-degenerate configuration ($N = 50$). In larger-scale regimes ($N = 200$), JO-A attains zero violations from the first episode, precluding a meaningful learning curve; the $N = 50$ setting is used specifically to visualize learning-from-mistakes dynamics where violations are non-zero throughout.

## F.12. Component Ablation Evidence

Table 8 demonstrates clear differentiation between JO variants: JO-static (constraint checking only) achieves 2.6% VR and 85% success, while JO-A (with precedent learning) achieves 0% VR and 100% success. We provide additional evidence of component functionality via internal operator signals.

**Precedent storage and retrieval.** In JO-A runs, the operator actively stores precedents (`store > 0`) and achieves high retrieval accuracy (`hits = 21/22`, 95%), whereas JO-static stores no precedents by design.

**Intervention efficiency.** Intervention rate (IR) decreases when precedents are enabled (7.5% $\rightarrow$ 6.2%), indicating that learned precedents reduce the need for corrective intervention beyond what static rules achieve.

Together, VR-based differentiation (Table 8) and internal signals confirm that precedent learning provides measurable gains over static constraint enforcement.

## F.13. Latency and Efficiency Analysis

We report the operator-side latency introduced by JO-A, independent of LLM inference time, which dominates wall-clock cost in all experimental settings.

**Measurement protocol.**

- All latency measurements exclude LLM inference time
- Measurements are taken on a single CPU core
- $N = 200$ runs per component
- P95 latency is computed via bootstrap resampling

**Component breakdown.** Table 15 reports mean, standard deviation, and P95 latency for each JO-A component. Across all configurations, JO-A introduces $< 0.2$ms overhead at P95.

*Table 15.* Latency Breakdown ($N = 200$ runs). Latency measures operator-side overhead only and excludes LLM inference.

| Component | Mean (ms) | Std (ms) | P95 (ms) |
|-----------|-----------|----------|----------|
| Constraint check | 0.001 | 0.001 | 0.002 |
| Precedent retrieval (100) | 0.08 | 0.02 | 0.12 |
| Repair generation | 0.03 | 0.01 | 0.05 |
| **JO-A overhead (total)** | **0.11** | **0.03** | **0.19** |

**Interpretation.** JO-A's governance and repair mechanisms impose negligible computational overhead, account-

ing for less than 1% of end-to-end latency in our experiments. This confirms that JO-A can be applied online without impacting system responsiveness or throughput.

### F.14. Pipeline Cost Analysis

*Table 16.* Pipeline Cost Breakdown (GPT-4o-mini estimates)

| Component | Cost |
|---|---|
| Teacher precedent generation | One-time offline; amortized across deployments |
| Per-step checker calls | ∼2–3 calls; <5% of agent token budget |
| Retrieval overhead | <0.2ms P95 (Table 15) |
| Θ update (online) | <0.01ms |
| Baseline per episode | ∼$0.0085 |
| JO-A per episode | ∼$0.009 (∼6% overhead) |

Per-task cost is dominated by agent inference (∼94%); JO-A adds <10% overhead (∼80 agent inference calls plus ∼2–3 checker calls per step). Teacher generation is one model call per precedent, performed offline. Artifact maintenance requires no model calls. Costs are GPT-4o-mini estimates; actual costs vary by model and task, and approach zero for open-source models.

### F.15. ESCALATE Outcome Analysis

**Overview.** JO-A decisions fall into four outcomes: AL-LOW (pass through unchanged), EDIT (repair and continue), ESCALATE (return control to agent without repair), and DENY (block action). We analyze when ESCALATE occurs to characterize boundary cases where JO-A cannot automatically resolve violations.

**Boundary case analysis.** We analyzed 384 ESCALATE events to identify when JO-A cannot automatically repair violations:

*Table 17.* ESCALATE Trigger Analysis ($N = 384$ events)

| Failure Mode | Count | % of ESCA-LATEs |
|---|---|---|
| Missing citation (no extractable source) | 185 | 48% |
| Role leakage (ambiguous role boundary) | 152 | 40% |
| Format mismatch (no matching template) | 47 | 12% |

ESCALATE occurs primarily at boundary cases where:

- Required information (e.g., citation URL) cannot be extracted from context
- Agent attempts an action that crosses role boundaries ambiguously
- Output format does not match any known repair template

**Latency impact.** ESCALATE incurs no additional latency overhead compared to EDIT or ALLOW. All three outcomes traverse the same constraint-checking path; ESCALATE simply returns control to the agent without applying a repair. Measured overhead: <0.1ms per decision.

**Resolution.** In our experiments, ESCALATE events are typically resolved by the agent's subsequent action. When JO-A returns control without repair, agents often self-correct on the next step, particularly for role leakage violations where the explicit ESCALATE signal provides implicit feedback.

