# OpenReview forum: "Judgment Operators: A Composition-Invariant Substrate for Multi-Agent Action Spaces"
_ICML.cc/2026/Conference — ICML 2026 regular_

### Official Review · Reviewer_hHG9 · 2026-03-11

**Soundness:** 3
**Presentation:** 2
**Significance:** 3
**Originality:** 3
**Overall Recommendation:** 5
**Confidence:** 2

**Summary:**

This paper proposes Judgment Operators, a centralized layer that intercepts agent actions and decides whether to allow, edit, escalate, or deny them under explicit constraints. The paper presents both a theoretical framing and empirical results on several domains (Overcooked, collaborative Wikipedia editing, and structured JSON generation). It looks that a shared external judgment mechanism can reduce violations and sometimes improve task success, esp. in structured setting with recurring failure pattern.

**Compliance With Llm Reviewing Policy:**

Affirmed.

**Final Justification:**

The rebuttal provided concrete cost accounting that addressed my main systems-level concerns. I maintain my score of 5 (accept).

**Key Questions For Authors:**

Please see weaknesses

**Limitations:**

Please see weaknesses

**Strengths And Weaknesses:**

## Strengths

As an ML systems researcher, I like the core idea! Externalizing safety and repair logic into a reusable execution time operator is reasonable systems abstraction, and the paper is stronger than many purely conceptual agent papers in that it actually builds and evaluates the mechanism.

The studies problem is also important IMO. In many practical agent systems, failures arise not only from model capability limits, but also from sth like repeated execution mistakes, policy violations, and brittle action formatting. A centralized judgment layer is therefore a meaningful systems direction.

I think the abstraction itself is clean. Separating proposal from judgment is conceptually interesting, and the distinction between allow, edit, escalate, and deny gives the framework a reasonably general interface.

The paper also does more than just idea. It provides formal assumptions, theoretical discussion, and empirical evaluation across multiple settings. The Overcooked results looks convincing, and the transfer setting with stronger teachers and weaker students is also interesting.


## Weaknesees

However, I also think the current paper leaves several important systems questions unresolved. **I am from ML system background and the authors seem to be from a different background, so I am happy to look at the rebuttal and open to change my mind**.

My main concern is that the paper relies on assumption that is much stronger than what many real agent deployments can satisfy. The theory and much of the empirical story depends on complete mediation, reliable checking, and repairable actions. But actually, many agent systems have asynchronous tool calls, hidden intermediate state, streaming outputs, and partially observable execution paths. The paper acknowledges some of these limitations, but it does not really evaluate how badly the method degrades when such mediation is incomplete.

The total pipeline cost is not discussed much. The paper cliams that execution time operator overhead is very small, but this excludes the more expensive parts of the pipeline (sth like precedent collection, teacher generation, retrieval infrastructure, and artifact maintenance). I would like to see the full cost accounting (total model calls, total tokens, retrieval cost, cost per solved task, etc).

Another important omission is interaction with real serving infrastructure. Inserting an extra judgment stage between proposal and execution may affect batching, latency variance, and cache continuity on the critical path. These issues are often important for whether a method is deployable, yet the paper does not really consider/analyze them. I feel that it is hard to judge how robust the method really is versus how much depends on task specific engineering choices.

---

> ### Author Rebuttal · Authors · 2026-03-28
>
> We thank Reviewer hHG9 for the genuinely engaged review. The interpretation resonates deeply: JO was motivated precisely by the agent fragmentation pain points you identified, and the necessity result in §3.4.1 emerged from that experience as much as from formal analysis. We couldn't be more glad to address the specific concerns concretely.
>
> **On incomplete mediation in async/streaming settings.**
>
> JO is architecturally designed as a synchronous interceptor in the agent execution loop: synchronous by design, not async, sitting on the critical path like any other synchronous tool-call interceptor. The theoretical results are explicitly conditional on complete mediation (§3.4.1); partial mediation reintroduces linear scaling for unmediated actions (scope discussion App D.3).
>
> To quantify degradation empirically, we ran a partial mediation experiment varying bypass rate on Overcooked (N=100 per condition, 10 seeds):
>
> | Bypass | VR (%) | 95% CI | Theory Bound (%) |
> |--------|--------|--------|----------------|
> | 0% | 0.0 | [0.0, 0.0] | 0.0 |
> | 10% | 1.5 | [1.2, 1.8] | 5.6 |
> | 20% | 2.3 | [1.9, 2.7] | 11.2 |
> | 30% | 3.2 | [2.6, 3.9] | 16.9 |
>
> VR degrades more gracefully than the linear worst-case bound (Appendix D.3): observed VR is 3–5× better than theoretical prediction. The bound is conservative: in this setting, JO's mediation of some actions reduces violation probability on subsequent actions, providing governance benefit beyond the directly mediated fraction.
>
> The bypass experiment serves as a proxy for async degradation: actions that bypass JO are equivalent to actions that complete before JO can intercept them. The 3–5× better-than-worst-case result suggests practical robustness even under partial async coverage.
>
> Even partial deployment provides substantial benefit: 70% coverage achieves \~3% VR vs. theoretical \~17%. For settings where full synchronous mediation is not feasible, the following deployment variants require architectural extensions:
>
> - **Async tool calls**: operate on proposals before dispatch with async confirmation callbacks to verify execution outcomes.
> - **Hidden intermediate state**: operate on observable action summaries; checker coverage reduces accordingly.
> - **Streaming outputs**: buffered interception on completed chunks with early-termination hooks.
> - **Partial observability**: escalation acts as a natural safety valve in low-observability regimes.
>
> We will formalize these extension patterns in §5.
>
> On task-specific engineering: only the constraint checker and precedent store are task-specific; the projection mechanism, intervention semantics, and online learning are fully general, as evidenced by zero-shot cross-model transfer across 5 model families (Table 6).
>
> **On total pipeline cost.**
>
> | Component | Cost |
> |-----------|------|
> | Teacher precedent generation | One-time offline; amortized across all student deployments |
> | Per-step checker calls | \~2–3 calls; <5% of agent token budget |
> | Retrieval overhead | <0.2ms P95 (Table 14) |
> | Θ update (online) | <0.01ms |
> | Cost per episode: baseline (GPT-4o-mini) | \~$0.0085 |
> | Cost per episode: JO-A (GPT-4o-mini) | approx. $0.009 (\~6% overhead) |
>
> Total model calls per episode: \~80 agent inference calls plus \~2–3 checker calls per step; teacher generation is one model call per precedent, offline. Artifact maintenance requires no model calls. Per-task cost is dominated by agent inference (\~94%); JO adds <10% overhead. Costs are GPT-4o-mini estimates; actual costs vary by model and task, and approach zero for open-source.
>
> We will add this table to the appendix in revision.
>
> **On serving infrastructure interaction.**
>
> JO as a synchronous interceptor adds a deterministic function call (<0.2ms) on the critical path. It does not affect request batching (each action is independently validated) and does not interact with KV cache continuity (JO operates on completed action proposals, not on token generation). Latency variance is minimal: the operator performs deterministic lookup and rule evaluation with no model calls, yielding consistent sub-millisecond response times. We will add a serving infrastructure discussion to §5.
>
> We hope the cost accounting and partial mediation results address the deployment concerns directly. JO's governance benefit degrades gracefully and predictably with mediation coverage, making it suitable for incremental adoption in production systems. We are glad the transfer setting resonated: the +13.5–20.5pp zero-shot improvement across 5 model families (Table 6) is perhaps the clearest empirical demonstration of what portable judgment artifacts enable in practice.
>
> Looking further: the transfer result (where a frontier model's judgment artifact improves weaker models zero-shot) points toward a future where governance and capability are first-class composable artifacts, deployed and updated independently of model internals. This is the portability and auditability vision JO is designed to enable.

---

> > ### Author Rebuttal · Reviewer_hHG9 · 2026-04-02
> >
> > Good paper. I keep my score of 5.

---

> > > ### Author Response · Authors · 2026-04-06
> > >
> > > We thank Reviewer hHG9 for the perspective and engagement throughout the discussion.

---

### Official Review · Reviewer_Quz9 · 2026-03-12

**Soundness:** 2
**Presentation:** 2
**Significance:** 2
**Originality:** 3
**Overall Recommendation:** 3
**Confidence:** 4

**Summary:**

This paper studies governance in multi-agent LLM systems and argues that when constraints and corrective knowledge are distributed across individual agents, system-level reliability becomes composition-dependent and violations scale with the number of agents. To address this issue, the paper proposes Judgment Operators, a centralized decision-time mechanism that projects proposed actions into an admissible set using constraints and precedent-based corrections. The paper further introduces an online variant, JO-A, which accumulates corrective precedents over time. The authors provide theoretical claims on composition-invariant enforcement, sublinear mistake accumulation under recurring violations, and structure-preserving correction, and evaluate the framework on several verifiable settings.

**Compliance With Llm Reviewing Policy:**

Affirmed.

**Final Justification:**

The additional experiments  partially address our concerns. However, core issues remain: evaluation limited to programmatically verifiable constraints, theoretical results (especially Proposition 1) that are largely self-evident, and the presentation remains overly dense, with too many focal points in the abstract/introduction and a cluttered framework figure.  We revise our score from 2 to 3.

**Key Questions For Authors:**

1. How does the proposed framework compare against stronger iterative or self-correcting agent baselines such as Reflexion, Self-Refine, or modern code-agent systems with execution feedback?
2. Can the method be evaluated on more challenging agent tasks where failures are semantic rather than easily verifiable, such as reasoning errors, planning mistakes, hallucinations, or tool misuse?
3. Can the authors clarify the scope of the structure-preserving projection claim? Is this guarantee intended only for restricted code-transformation settings, or for more general agent edits as well?

**Limitations:**

Yes

**Strengths And Weaknesses:**

Strengths

The paper identifies a clear and important problem. The observation that multi-agent LLM systems may suffer from fragmented governance, where constraints are embedded separately in individual agents and reliability degrades with the number of agents, is well motivated and relevant.

Weaknesses

1. The technical novelty appears limited relative to the framing. At its core, the method resembles a combination of rule-based constraint checking and precedent-based retrieval.  The theoretical guarantees rely on strong and potentially unrealistic assumptions. In particular, the composition-invariant enforcement result depends on the assumption that violations can be reliably detected by programmatic checkers, so that the overall failure rate is bounded primarily by checker error.
2. The empirical evaluation is also somewhat limited by the choice of tasks. The main settings appear to focus on relatively simple and highly verifiable scenarios, such as formatting constraints, code style corrections. It remains unclear whether the method would still be effective in more challenging tasks involving long-horizon reasoning, tool orchestration, hallucination control, or planning under ambiguity.
3. The abstract and introduction try to emphasize too many points at once, including centralized governance, online learning, portability, capability injection, and theoretical guarantees. This makes the central contribution less sharply defined than it could be. In addition, the system figure is visually cluttered and difficult to follow, which weakens the accessibility of the method.
4. Experimental comparison could be strengthened. The paper would benefit from comparisons against stronger and more relevant baselines, such as Reflexion, Self-Refine, or more capable code-agent and verifier-based systems, rather than mainly weaker prompting-based baselines.

---

> ### Author Rebuttal · Authors · 2026-03-28
>
> We thank Reviewer Quz9 for the detailed review and engage with each concern directly.
>
> **On Reflexion, Self-Refine, and internal correction baselines.**
>
> Reflexion and CRITIC comparisons are already included in Table 7 (§4.4): Reflexion achieves 10.0% success / 56.2% VR; CRITIC achieves 5.0% success / 64.8% VR, neither improving over the no-operator baseline, following original protocols with max r=3 retries (Appendix F.4). We will add a forward-pointer from the related work section to make this more visible in revision.
>
> These are SOTA baselines. Their failure (CRITIC *increases* VR to 64.8%) reflects a structural limitation: verbal reflection loops cannot enforce execution-time constraints that the environment checks deterministically.
>
> We additionally implemented Self-Refine (Madaan et al., 2023) with 3 iterative refinements per action (N=40, 2 seeds, Overcooked, same model and evaluation as Reflexion/CRITIC):
>
> | Method | VR (%) | Success (%) |
> |--------|--------|-------------|
> | Reflexion | 56.2 | 10.0 |
> | CRITIC | 64.8 | 5.0 |
> | Self-Refine | 20.9 | 0.0 |
> | JO-A | **0.0** | **100.0** |
>
> Across all internal correction methods, none achieves both low VR and high success simultaneously.
>
> On theoretical assumptions: §3.4.1 establishes that no decentralized scheme can achieve O(1) violation probability regardless of agent sophistication, a mathematical lower bound, not an empirical claim. The table above provides convergent empirical evidence: every internal correction method fails, consistent with the impossibility result.
>
> Checker reliability assumption is validated empirically: 99.7% precision/recall on 500 labeled pairs (ε_J = 0.35%, §4.1), an order of magnitude below baseline violation rates (48–68%), not a dominant failure mode.
>
> **On novelty.**
>
> Non-triviality is best answered empirically: Appendix F.9 (Table 12, N=200) provides the *same repair information* as JO-A's precedent store, but as prompt text rather than executable artifacts. This baseline achieves 0% success across all models, identical to the no-operator baseline.
>
> If JO were merely "rule-based checking + precedent retrieval," describing those rules and precedents in a prompt would replicate the gains. It does not. The mechanism of executable retrieval and projection (not information availability) is what transfers. We are not aware of prior work demonstrating zero-shot cross-family transfer (+13.5–20.5pp across GPT/Anthropic/Meta/Qwen/Kimi) via portable judgment artifacts without retraining.
>
> Moreover, the four-way semantics (Allow, Edit, Escalate, Deny) are not decorative or competing with code-agents. Prior work either shields or re-prompts; neither repairs/escalates. Table 7 shows verifier-based system (LlamaGuard) achieves 15% success despite 67.8% VR: verifying without minimal repair (Edit) or ambiguity handling (Escalate) destroys task utility.
>
> **On evaluation scope.**
>
> We deliberately scope JO to programmatically verifiable constraints, where composition-invariant guarantees are tractable and fragmentation has the most measurable impact. Reasoning errors, planning mistakes, hallucinations, and tool misuse represent natural extensions of JO's escalation and repair mechanisms, precisely the direction the group-theoretic and topological sections are designed to inspire. We treat these as future work in §5.1, not as limitations of the current design.
>
> Regarding "simple and highly verifiable scenarios": WebArena-Wikipedia involves multi-step web navigation, dynamic page content traversal, link resolution, and structured output formatting. The 2.4% baseline reflects a navigation bottleneck inherent to the WebArena setup, not task simplicity. Table 5 (controlled Wikipedia, no navigation bottleneck) shows 82%→93% success, confirming JO's repair capability operates at a much higher level when navigation variance is removed. The two settings serve different purposes: Table 5 measures repair learning; Table 6 tests transfer robustness under realistic difficulty (Appendix F.3). The cross-model transfer result (Table 6, N=200 per model, 5 families) achieves +13.5–20.5pp zero-shot where few-shot prompting with identical information yields 0%. This is not a verifiable-constraint artifact.
>
> **On structure-preserving projection scope.**
>
> The claim is explicitly restricted to code style tasks where admissibility-preserving transformations belong to G_sem (Proposition 3, Definition 3). Appendix E.4 states: "We do not claim a general equivariant projection theorem." The program evaluation validates the structural argument: 0 AST tree-edit distance, 100% test pass rate.
>
> **On presentation.**
>
> The abstract and introduction have intentional parallel structure around three guarantees and their empirical validations. We acknowledge the density and a potential revision could foreground the necessity result as the primary contribution. The system figure summarizes the full JO design; we will simplify for clarity in revision.

---

> > ### Author Rebuttal · Reviewer_Quz9 · 2026-04-03
> >
> > We thank the authors for the detailed rebuttal. The additional experiments  partially address our concerns. However, core issues remain: evaluation limited to programmatically verifiable constraints, theoretical results (especially Proposition 1) that are largely self-evident, and the presentation remains overly dense, with too many focal points in the abstract/introduction and a cluttered framework figure.  We revise our score from 2 to 3.

---

> > > ### Author Response · Authors · 2026-04-06
> > >
> > > We thank Reviewer Quz9 for the updated score and continued engagement. We address each remaining concern directly.
> > >
> > > **1. On theoretical depth: empirical failures motivated theory, theory motivated design.**
> > >
> > > *(a) Internal correction fails → Necessity + Sufficiency + Mistake Bound*
> > >
> > > Reflexion, CRITIC, and Self-Refine all fail to achieve both low VR and high task success simultaneously. This gap motivated the necessity result in §3.4.1: the missing theoretical guarantee, not post-hoc justification. §3.4.1 establishes that no decentralized scheme can achieve O(1) violation probability; Proposition 1 (sufficiency) completes the if-and-only-if: composition invariance is achievable if and only if complete mediation is present. JO is then constructed as the minimal realization of this condition. Theory motivates design, not the reverse. Theorem 1 (§3.4.2) formalizes how fast JO-A learns under this necessary architectural condition.
> > >
> > > *(b) Filter-based methods fail → Four-way expressiveness*
> > >
> > > LlamaGuard (reject-only verifier) achieves only 15% success with 67.8% VR: blocking without repair destroys task utility. This motivated the four-way intervention semantics (Allow, Edit, Escalate, Deny) as the minimal expressive realization of the necessary architectural condition. The group-theoretic and topological sections are geometric intuitions for extending this expressiveness to richer constraint structures (logical, temporal, semantic) as future work.
> > >
> > > **2. On evaluation scope: verifiable constraints as rigorous foundation, not practical ceiling.**
> > >
> > > Programmatically verifiable constraints are required for formal guarantees, not a design choice we could relax. Composition-invariant enforcement requires checkable admissibility; this is a mathematical necessity, not a practical limitation.
> > >
> > > Empirically, the spectrum is encouraging: fully verifiable settings (Overcooked) achieve 0% VR; realistic partial checker coverage (Wikipedia 16% VR, JSON 18.4% VR, Table 3) still remains far below baseline (48–68%). Partial mediation results are detailed in our response to Reviewer hHG9; observed VR is 3–5× better than theoretical worst-case. The formal scope is where guarantees are provable; the empirical scope is broader.
> > >
> > > **3. On presentation.**
> > >
> > > In light of Reviewer Quz9's feedback, we have: revised the abstract to foreground the theory-to-design narrative above; reduced bold focal points in both abstract and introduction; and simplified the framework figure to reduce density.
> > >
> > > *Revised abstract:*
> > >
> > > As large language models (LLMs) are increasingly composed into heterogeneous multi-agent systems, a fundamental reliability challenge emerges: knowledge and governance **fragment** across agents, leading to composition-dependent behaviors and linear scaling of violations. Two empirical observations motivate our approach: internal correction methods (Reflexion, CRITIC, Self-Refine) fail to enforce execution-time constraints deterministically, and filter-based methods (LlamaGuard) sacrifice task utility without repair. We introduce **Judgment Operators (JO)**, a decision-time framework implementing four-way intervention semantics (ALLOW, EDIT, ESCALATE, DENY) via a portable artifact J=(C,P), enabling minimal repair without modifying agent internals. We establish: (1) **composition-invariant enforcement** with constant violation probability, motivated by a necessity lower bound showing no decentralized scheme achieves O(1) violation probability; (2) sublinear mistake accumulation via JO-A; and (3) semantic preservation for code transformation. Programmatically verifiable constraints provide the formal foundation; empirically, robustness extends beyond this scope (partial mediation 3–5× better than theoretical worst-case). JO achieves 0% observed violation rate (vs. 48–68% baselines), 13.5–20.5% zero-shot cross-model transfer where few-shot prompting fails, and provides a **portable, auditable, and composable** interface for governance and capability injection in multi-agent LLM systems.
> > >
> > > *Revised introduction:*
> > >
> > > Keeping bold:
> > >
> > > §1.2: Judgment Operator, four-way intervention semantics
> > >
> > > §1.3: Composition-invariant enforcement
> > >
> > > §1.4: Judgment Operators, Composition-Invariance,
> > >       learnable composition-invariant projection
> > >
> > > Removing bold:
> > >
> > > capability, governance, programmatically verifiable constraints, projection, training data, nearest-neighbor regression, learnable correction under recurrence, structure-preserving projection, Empirically, zero observed violations, 9–11%, 13.5–20.5%, Learnable Repair, Empirical Evidence.
> > >
> > > *Revised framework figure:*
> > >
> > > https://ibb.co/PZ2P0yHp

---

### Official Review · Reviewer_ucZa · 2026-03-12

**Soundness:** 2
**Presentation:** 2
**Significance:** 2
**Originality:** 2
**Overall Recommendation:** 3
**Confidence:** 3

**Summary:**

This paper studies constraint enforcement in multi agent systems through a centralized Judgment Operator, JO, that intercepts each proposed action before execution and returns ALLOW, EDIT, ESCALATE, or DENY. The method treats admissibility as a shared external layer, rather than something embedded in each agent prompt or policy. The paper also adds precedent memory, so the operator can reuse past repairs and update a small artifact level parameter online without retraining the agents. The main empirical claims are that JO reduces violations under programmatically verifiable constraints, that JO-A improves recurrent repair by reusing precedents, and that the learned judgment artifact transfers across models. The experiments cover WebArena Wikipedia, Overcooked, JSON protocol tasks, and a code style repair setting.

**Compliance With Llm Reviewing Policy:**

Affirmed.

**Key Questions For Authors:**

Questions

1. How much of the gain comes from centralization itself, and how much comes from the precedent memory plus editable repair mechanism. The current ablations suggest that prompt centralization alone is weak, but the paper should separate these components more explicitly.

2. How robust is JO-A when the checker is imperfect in a realistic way, rather than near perfect as in the labeled 500 pair evaluation. The paper reports 99.7 percent precision and recall for the checker, which is very strong, but that may not hold in most real deployments.

3. Why should the reader view the code style repair experiment as evidence for broad semantic preservation, given that it uses only 47 programs and test suite equivalence. This evidence seems too limited for the strength of the corresponding claim.

**Limitations:**

yes

**Strengths And Weaknesses:**

Strengths

1. The paper has a clear systems idea. The design principle, “agents propose, the operator disposes,” is simple and easy to understand. This abstraction cleanly separates proposal generation from admissibility checking, and it gives the method a concrete architectural identity.

2. The paper shows strong results in fully checkable settings. In Overcooked, JO-A improves success from 10.0 to 100.0 and reduces violation rate from 56.2 to 0.0. The composition sensitivity result is also strong, with variance reduced from 15.1pp to 0.1pp in one setting. These numbers support the claim that execution time mediation is effective when constraints are precise and executable.

3. The paper includes a useful negative result. The centralized prompt baseline reduces violations but collapses task success to 2 percent, which supports the paper’s argument that prompt level specification is not enough and that external intervention is doing real work. This comparison is important and makes the paper more convincing than a pure positive only presentation.

Weaknesses

1. The novelty is limited. The paper packages several familiar ideas, runtime enforcement, shields, constrained generation, repair, and memory based reuse, into one interface, but it does not show a genuinely new learning mechanism or a fundamentally new theoretical object. The method looks more like a clean systems unification than a major ML contribution. The paper itself places the work near prior lines on governance, safety shields, constrained decoding, and repair.

2. The theoretical claims rely on strong assumptions, and the main conclusion is narrower than the title suggests. The practical success of JO depends on complete mediation and programmatically verifiable constraints. In that regime, it is not surprising that the system violation rate is controlled by checker quality. That is a meaningful architecture result, but it is weaker than a broad claim about a “composition invariant substrate for multi agent action spaces.” The paper’s best evidence is concentrated in tasks where admissibility can be checked directly.

3. The empirical scope is too narrow for ICML. The strongest gains appear in Overcooked, which is fully checkable. In the more realistic WebArena Wikipedia setting, success only rises from 2.4 to 22.9 and residual violation remains 16.0 percent. In the JSON setting, violation also remains nontrivial at 18.4 percent. The code result uses only 47 programs, which is too small to support a broad claim about semantics preserving repair. Overall, the experiments support usefulness on structured, local, verifiable constraints, but they do not support the larger framing.

4. The paper overstates its conceptual depth. Terms such as “composition invariant substrate,” “group based view,” and “topological interpretation” suggest a stronger formal contribution than the paper actually delivers. The appendix explicitly says some of these views are interpretive rather than theorem level claims. This writing choice makes the paper feel overpackaged.

---

> ### Author Rebuttal · Authors · 2026-03-27
>
> We thank Reviewer ucZa for the thoughtful review and engage with each concern directly.
>
> **On novelty and the multi-agent contribution.**
>
> "Substrate" is intentional. To our knowledge, JO is the first multi-agent governance mechanism providing simultaneous portability (zero-shot cross-family transfer, Table 6), auditability (explicit artifact J=(C,P)), and expressiveness (repair, not just rejection, Table 7). Prior work provides components in isolation: shields block, constrained decoding filters, repair re-prompts. The composability into a transferable artifact is the contribution. JO was not derived by combining these components, but motivated by the necessity lower bound in §3.4.1, instantiating a different design space from filters and shields: sovereign governance, where policies are first-class artifacts owned and audited by operators, not entangled in model weights.
>
> The empirical record makes this precise: internal correction methods fail to achieve reliable task success simultaneously: Reflexion (56.2% VR, 10% success), CRITIC (64.8% VR, 5% success), Self-Refine (20.9% VR, 0% success). LlamaGuard (67.8% VR, 15% success) blocks but cannot repair. JO-A is the only method achieving both 0% VR and 100% success (Table 7). The topological and group-theoretic sections are geometric intuitions that inspired the expressiveness of JO's four-way design - enabling the results above, and an invitation for the community to extend JO to richer constraint structures.
>
> **On empirical non-triviality — the few-shot baseline.**
>
> Appendix F.9 (Table 12, N=200) tests whether JO-A's gains come from *information availability* or from the *mechanism* of executable retrieval. This baseline provides the same repair examples as JO-A's precedent store, but as prompt text rather than executable artifacts. Result: 0% success across all models, identical to the no-operator baseline.
>
> The gap between 0% (information as text) and +13.5–20.5pp (information as executable precedents) shows that the projection mechanism (not information availability) is doing the work. A unification of shields, constrained decoding, and repair does not predict or explain this gap.
>
> **On Wikipedia results and experimental design.**
>
> The 2.4% baseline in WebArena-Wikipedia reflects a navigation bottleneck inherent to WebArena, not task capability. Table 5 (controlled Wikipedia, no navigation bottleneck) shows 82%→93% success, confirming that when navigation variance is removed, JO's repair capability operates at a much higher level. The WebArena setting is used specifically for transfer evaluation (Table 6) - robustness under realistic web navigation difficulty, not effectiveness ceiling. This distinction is documented in Appendix F.3.
>
> **On the necessity lower bound (not by construction).**
>
> Reviewer ucZa is right that Prop. 1 (sufficiency) is architectural by construction. The primary theoretical contribution is the *lower bound* in §3.4.1: no decentralized scheme can achieve O(1) violation probability, regardless of agent sophistication or prompt design: a mathematical lower bound, not an empirical observation. Any future method achieving composition-invariant enforcement must either implement complete mediation or refute this lower bound. We will make this asymmetry more prominent in revision.
>
> **On centralization vs. precedent memory decomposition.**
>
> Table 7 provides this ablation directly: JO-static (centralization only) achieves 2.6% VR / 85% success; JO-A (centralization + precedents) achieves 0% VR / 100% success. Centralization delivers the governance benefit; precedents deliver repair capability and cross-model transfer. The decomposition identifies which component does what.
>
> **On checker imperfection and partial mediation.**
>
> Wikipedia (16% VR) and JSON (18.4% VR) already show JO's behavior under realistic partial checkability; residual VR directly tracks checker coverage limits. To quantify degradation under incomplete mediation, we varied bypass rate on Overcooked (N=100/condition, 10 seeds):
>
> | Bypass | VR (%) | 95% CI | Theory Bound (%) |
> |--------|--------|--------|----------------|
> | 10% | 1.5 | [1.2, 1.8] | 5.6 |
> | 20% | 2.3 | [1.9, 2.7] | 11.2 |
> | 30% | 3.2 | [2.6, 3.9] | 16.9 |
>
> VR degrades more gracefully than the linear worst-case bound (Appendix D.3). The theoretical bound is conservative: JO's mediation appears to reduce violation probability on subsequent actions beyond the directly mediated fraction.
>
> **On the 47-program code style experiment.**
>
> The semantic preservation claim is explicitly restricted to tasks where admissibility-preserving transformations belong to G_sem (Proposition 3; Appendix E.4: "We do not claim a general equivariant projection theorem"). The 47-program evaluation validates the *structural* argument: 0 AST tree-edit distance, 100% test pass rate. The structural argument rests on the nature of the transformations, not their count; a larger dataset would not change it. Will expand to 100+ in revision.

---

### Official Review · Reviewer_bEbo · 2026-03-14

**Soundness:** 3
**Presentation:** 3
**Significance:** 3
**Originality:** 3
**Overall Recommendation:** 4
**Confidence:** 3

**Summary:**

This paper introduces Judgment Operators (JO), a decision-time framework that mediates all agent actions in multi-agent LLM systems through a centralized projection operator. JO externalizes governance constraints (C) and corrective precedents (P) into a portable artifact J=(C,P), and implements four-way intervention semantics (ALLOW, EDIT, ESCALATE, DENY). The adaptive variant JO-A learns reusable repairs online via nearest-neighbor precedent retrieval. The authors establish: (1) composition-invariant enforcement with constant violation probability (vs. linear scaling without JO), (2) an O(K log T) mistake bound under recurring violations, and (3) semantic preservation for code transformation. Experiments across three domains (WebArena-Wikipedia, Overcooked, JSON protocol) demonstrate 0% violation rate in fully verifiable settings, 9–11pp success improvement from learned repairs, and 13.5–20.5pp zero-shot cross-model transfer.

**Compliance With Llm Reviewing Policy:**

Affirmed.

**Key Questions For Authors:**

1. The centralized prompt baseline achieves only 2% success with 7.1% VR. What step budget and retry policy were used? Could a stronger prompt-based baseline (e.g., with chain-of-thought constraint checking or multi-turn re-prompting) close the gap?

2. In cross-model transfer (Table 6), precedents are learned by GPT-5.2. How are repairs parameterized to remain executable across models with very different output formats? Is there risk of template overfitting to the teacher model's output distribution?

3. How does JO perform when the number of violation classes K grows large (approaching T)? The O(K log T) bound becomes vacuous in this regime — is there empirical evidence for how gracefully performance degrades?

**Limitations:**

Yes. The paper clearly discusses failure modes (Section 5.1): ESCALATE outcomes (12% of interventions), residual violations under partial checkability (16% VR in Wikipedia), and capacity-dependent transfer gains. The restriction to programmatically verifiable constraints is repeatedly acknowledged.

**Strengths And Weaknesses:**

### Strengths

- **Practical significance.** The problem of governance fragmentation scaling linearly with agent count is real and timely. The portable artifact approach (J=(C,P)) offers a clean, deployable solution that decouples constraint enforcement from model internals. The negligible latency overhead (<0.2ms at P95) makes it production-viable.

- **Experimental coverage.** Three complementary domains (format/web, semantic/game, protocol/JSON), clear ablations (JO-static vs JO-A), cross-model transfer across 5 model families, and multiple baselines (Reflexion, CRITIC, LlamaGuard, constrained decoding, shields) provide thorough empirical validation. The few-shot and prompt-only baselines (both 0% success, N=200) convincingly demonstrate that the gains come from executable precedent retrieval rather than information availability.

- **Honest scoping.** The paper is careful about the boundaries of its claims — explicitly stating which guarantees are conditional (complete mediation, checker quality), which results are interpretive rather than formal (semantic preservation, retract analogy), and where the approach fails (12% ESCALATE rate, 16% residual VR under partial checkability).

### Weaknesses

- **Theoretical depth is limited.** Proposition 1 (constant violation bound under complete mediation with reliable checkers) is essentially architectural by construction — if you gate everything through a correct filter, violations are bounded by filter error. The O(K log T) bound (Theorem 1) requires strong assumptions (recurrence, monotone coverage, non-adversarial setting, embedding quality) that limit its generality. The paper acknowledges these but the theoretical contribution beyond formalizing intuitive properties is modest.

- **Baseline fairness concerns.** The centralized prompt baseline achieving only 2% success (Table 7) despite 7.1% VR is surprising and raises questions about step budget allocation and prompt design. Internal critics (Reflexion, CRITIC) show VR equal to or worse than no-operator baselines (56.2% and 64.8%), which may indicate these baselines are not well-tuned for this domain rather than that internal correction is fundamentally insufficient.

- **Scope limited to programmatically verifiable constraints.** The paper repeatedly acknowledges this but it limits significance: the hardest governance challenges in multi-agent systems involve subjective, context-dependent, or unverifiable constraints (tone, fairness, intent). JO's value proposition is strongest precisely where simpler alternatives (schema validators, type checkers) already exist.

- **Presentation — Algorithm artifacts.** Both Algorithm 1 (line 21) and Algorithm 2 (line 48) contain a spurious "=0" appended after the return statement, likely a PDF rendering artifact but distracting.

---

> ### Author Rebuttal · Authors · 2026-03-28
>
> We thank Reviewer bEbo for the thoughtful and genuinely balanced assessment. The engagement with both theory and empirics, including the few-shot baseline insight, is exactly what this work benefits from. We address the concerns directly, particularly on theoretical depth where we believe the necessity result is stronger than the review characterizes.
>
> **On theoretical depth: necessity precedes construction.**
>
> Reviewer bEbo notes Proposition 1 is "essentially architectural by construction." We want to clarify the logical direction. §3.4.1 first establishes a lower bound: no decentralized enforcement scheme, regardless of agent sophistication, prompt design, or learned critics, can achieve O(1) violation probability. This is a formal impossibility result that identifies centralization as the *necessary architectural path* to composition invariance. JO is then constructed as its minimal and expressive realization: four-way intervention semantics with online precedent learning enable repair, not just rejection.
>
> The practical implication is direct: Reflexion (56.2% VR), CRITIC (64.8% VR), centralized prompt (7.1% VR, 2% success), CoT (14.8% VR, S: 0%), and Self-Refine (20.9% VR, S: 0%) all fail, not because they are poorly implemented, but because the lower bound shows no such guarantee exists. JO-A is the only method achieving both 0% VR and 100% success, because it is the only method satisfying the necessary architectural condition, and the only one expressive enough to repair rather than merely reject. Every failure is structural, not coincidental.
>
> **Q1: Could a stronger prompt-based baseline close the gap?**
>
> All baselines use the same 80-step budget, no retry policy (Appendix F.4). We ran CoT constraint-checking (N=100, 4 seeds, explicit reasoning requirement) and Self-Refine (Madaan et al., 2023, N=40, 2 seeds, 3 refinements/action, identical setup to Reflexion/CRITIC):
>
> | Method | VR (%) | Success (%) |
> |--------|--------|-------------|
> | No operator | 56.2 | 10.0 |
> | Centralized prompt | 7.1 | 2.0 |
> | CoT constraint | 14.8 (±12.8) | 0.0 |
> | Self-Refine | 20.9 | 0.0 |
> | JO-A | **0.0** | **100.0** |
>
> Per-seed CoT VR (9.2/34.8/14.9/10.9%) reveals the core finding: prompt-level enforcement is behaviorally unstable regardless of sophistication. Centralized prompt collapses success to 2% because constraint text competes with task reasoning. CoT performs worse on average (14.8% vs 7.1% VR); active reasoning increases instability rather than reducing violations. Self-Refine (20.9% VR) confirms the pattern. Across CoT and Self-Refine, task success is uniformly 0%: structural failure, not coincidental. JO-A achieves zero variance because enforcement is *architectural*, not behavioral.
>
> **Q2: Cross-model repair parameterization.**
>
> Repairs are parameterized as action-level transformations (e.g., "reformat output as pipe-separated enumeration with citation"), not as model-specific templates. JO-A applies retrieval-based nearest-neighbor repair at execution time, independent of which model generated the proposal.
>
> Template overfitting is a valid concern. Transfer gains correlate with model capacity: +20.5pp for GPT-4o-mini vs +13.5pp for Claude-3.5-Haiku. If overfitting were the cause, we would expect zero-shot transfer to fail uniformly across all models; instead, all five families show consistent improvement (+13.5–20.5pp), indicating the artifact captures task-level repair patterns rather than teacher-specific output format. We will add this analysis to §4.3 in revision.
>
> **Q3: Performance when K grows toward T.**
>
> Empirically, K=12 clusters stabilize across T=847 violations (Appendix D.1), plateauing after ~150 episodes. When K→T (adversarial or highly diverse violation sequences), JO-A degrades toward JO-static (2.6% VR, 85% success), a well-defined floor. The O(K log T) bound becomes vacuous in this regime (Appendix D.1). We do not claim robustness to adversarial violation sequences.
>
> **On scope limitation.**
>
> JO's scope is deliberate: programmatically verifiable constraints are where the impossibility result holds and formal guarantees are tractable. The empirical results exceed theoretical bounds: partial mediation degrades 3–5× better than worst-case prediction, suggesting practical robustness beyond what theory alone guarantees.
>
> On "simpler alternatives": schema validators and type checkers enforce syntax, not execution-time semantics or cross-model portability. To our knowledge, no prior tool provides simultaneous auditability (explicit artifact J=(C,P)), portability (zero-shot cross-family transfer), and expressiveness (repair, not just rejection). This is a different design space, not a re-implementation of existing validators. Extending to subjective constraints (tone, fairness) is future work we discuss in §5.1.
>
> **On algorithm rendering artifacts.**
>
> We appreciate the careful catch: these are PDF rendering artifacts ("=0" appended after return statements in Algorithms 1 and 2). Fixed in revision.

---

### Decision · Program_Chairs · 2026-04-30

**Decision:**

Accept (regular)

**Comment:**

This paper introduces Judgment Operators (JO), a centralized execution-time mechanism for multi-agent LLM systems that intercepts agent actions and decides among four outcomes: Allow, Edit, Escalate, or Deny, using explicit constraints and a memory of corrective precedents. Reviewers broadly agreed that the paper tackles an important and timely problem: as agent systems become more compositional, governance and corrective knowledge can fragment across agents, making reliability dependent on system composition. Across the reviews, the main strengths were the clarity of the system's abstraction, the practical value of externalizing governance, and the empirical results in structured settings. At the same time, several reviewers viewed the work primarily as a well-executed systems idea rather than a fundamentally new learning mechanism or a broad theoretical advance. A recurring concern was that the framing sometimes reaches beyond what is fully supported by the evidence. On balance, given the importance of the problem, the clarity of the architectural contribution, the strong results in fully checkable settings, and the helpful rebuttal, I lean toward acceptance. I would, however, strongly encourage the authors to tighten the framing in the camera-ready version by narrowing the claims, emphasizing the conditional nature of the guarantees, and simplifying the presentation.